# Temporal evolution of single-cell transcriptomes of *Drosophila* olfactory projection neurons

Qijing Xie[1,2], Maria Brbic[3], Felix Horns[4,5], Sai Saroja Kolluru[4], Robert C Jones[4], Jiefu Li[1], Anay R Reddy[1], Anthony Xie[1], Sayeh Kohani[1], Zhuoran Li[1], Colleen N McLaughlin[1], Tongchao Li[1], Chuanyun Xu[1], David Vacek[1], David J Luginbuhl[1], Jure Leskovec[3], Stephen R Quake[4,6,7]*, Liqun Luo[1]*, Hongjie Li[1]

[1]Department of Biology, Howard Hughes Medical Institute, Stanford University, Stanford, United States; [2]Neurosciences Graduate Program, Stanford University, Stanford, United States; [3]Department of Computer Science, Stanford University, Stanford, United States; [4]Department of Bioengineering, Stanford University, Stanford, United States; [5]Biophysics Graduate Program, Stanford University, Stanford, United States; [6]Department of Applied Physics, Stanford University, Stanford, United States; [7]Chan Zuckerberg Biohub, Stanford, United States

**Abstract** Neurons undergo substantial morphological and functional changes during development to form precise synaptic connections and acquire specific physiological properties. What are the underlying transcriptomic bases? Here, we obtained the single-cell transcriptomes of *Drosophila* olfactory projection neurons (PNs) at four developmental stages. We decoded the identity of 21 transcriptomic clusters corresponding to 20 PN types and developed methods to match transcriptomic clusters representing the same PN type across development. We discovered that PN transcriptomes reflect unique biological processes unfolding at each stage—neurite growth and pruning during metamorphosis at an early pupal stage; peaked transcriptomic diversity during olfactory circuit assembly at mid-pupal stages; and neuronal signaling in adults. At early developmental stages, PN types with adjacent birth order share similar transcriptomes. Together, our work reveals principles of cellular diversity during brain development and provides a resource for future studies of neural development in PNs and other neuronal types.

*For correspondence:
steve@quake-lab.org (SRQ);
lluo@stanford.edu (LL)

Competing interests: The authors declare that no competing interests exist.

## Introduction

Cell-type diversity and connection specificity between neurons are the basis of information processing underlying all nervous system functions. The precise assembly of neural circuits involves multiple highly regulated steps. First, neurons are born from their progenitors and acquire unique fates through a combination of (1) intrinsic mechanisms, such as lineage, birth order, and birth timing; (2) extrinsic mechanisms, such as lateral inhibition and extracellular induction; and (3) developmental stochasticity in some cases (*Jan and Jan, 1994*; *Johnston and Desplan, 2010*; *Kohwi and Doe, 2013*; *Holguera and Desplan, 2018*; *Li et al., 2018*). During wiring, neurons extend their neurites to a coarse targeting region, elaborate their terminal structures, select pre- and post-synaptic partners, and finally form synaptic connections (*Sanes and Yamagata, 2009*; *Jan and Jan, 2010*; *Kolodkin and Tessier-Lavigne, 2011*; *Luo, 2020*; *Sanes and Zipursky, 2020*). Studies from the past few decades have uncovered many molecules and mechanisms that regulate each of these developmental processes.

The development of *Drosophila* olfactory projection neurons (PNs) has been extensively studied (*Jefferis et al., 2004*; *Hong and Luo, 2014*). PNs are the second-order olfactory neurons that receive organized input from olfactory receptor neurons (ORNs) at ~50 stereotyped and individually identifiable glomeruli in the antennal lobe, and carry olfactory information to higher brain centers (*Vosshall and Stocker, 2007*; *Wilson, 2013*; *Figure 1A*). Different types of PNs send their dendrites to a single glomerulus or multiple glomeruli (*Marin et al., 2002*; *Lai et al., 2008*; *Yu et al., 2010*; *Tanaka et al., 2012*; *Bates et al., 2020*). PNs are derived from three separate neuroblast lineages—anterodorsal, lateral, and ventral lineages, corresponding to their cell bodies' positions relative to the antennal lobe (*Jefferis et al., 2001*). PNs produced from the anterodorsal and lateral lineages (adPNs and lPNs) are cholinergic excitatory neurons. The fate of uniglomerular excitatory PN types, defined by their glomerular targets, is predetermined by their lineage and birth order (*Jefferis et al., 2001*; *Marin et al., 2005*; *Yu et al., 2010*; *Lin et al., 2012*). PNs produced from the ventral lineage (vPNs), on the other hand, are GABAergic inhibitory neurons (*Jefferis et al., 2007*; *Liang et al., 2013*; *Parnas et al., 2013*). The connectivity and physiology of PNs have also been systematically studied (*Bhandawat et al., 2007*; *Jeanne et al., 2018*; *Bates et al., 2020*).

Despite the fact that PNs are among the most well-characterized cell types in all nervous systems, their transcriptome-wide gene expression changes across different developmental stages with cell-type specificity are still unknown. This information can help us obtain a more complete picture of both known and unexplored pathways underlying neural development and function. Recently, the advent of single-cell RNA sequencing (scRNA-seq) has paved the way toward obtaining such data (*Li et al., 2017*; *Kalish et al., 2018*; *Zhong et al., 2018*; *Li, 2020*). Here, we profiled and analyzed the single-cell transcriptomes of most uniglomerular excitatory PNs. We identified the correspondence between two-thirds of transcriptomes and PN types at one stage and developed methods to reliably match transcriptomic clusters corresponding to the same types of PNs across different stages. We discovered that PN transcriptomes exhibit unique characteristics at different stages, including birth-order, neurite pruning, wiring specificity, and neuronal signaling. The identification of many differentially expressed genes among different PN types, such as transcription factors, cell-surface molecules, ion channels, and neurotransmitter receptors, provides a rich resource for further investigations of the development and function of the olfactory system.

## Results

### Single-cell transcriptomic profiling of *Drosophila* PNs at four developmental stages

The development of PNs follows the coordinated steps as previously described (*Hong and Luo, 2014*). Eighteen out of 40 types of adPNs are born embryonically and participate in the larval olfactory system. Then, during the larval stage, the rest of adPNs and all lPNs are born (*Jefferis et al., 2001*; *Marin et al., 2005*; *Yu et al., 2010*; *Lin et al., 2012*). During early metamorphosis following puparium formation, embryonically born PNs first prune terminal branches of dendrites and axons, and then re-extend their dendrites into the future adult antennal lobe, and axons into the mushroom body and lateral horn following the neurites of larval-born PNs (*Marin et al., 2005*). From 0 to 24 hr after puparium formation (APF), PNs extend their dendrites into the developing antennal lobe and occupy restricted regions. ORN axons begin to invade antennal lobe at ~24 hr APF. PN dendrites and ORN axons then match with their respective partners beginning at ~30 hr APF and establish discrete glomerular compartments at ~48 hr APF. Thereafter, they expand their terminal branches, build synaptic connections, and finally form mature adult olfactory circuits (*Jefferis et al., 2004*; *Figure 1B*).

To better understand the molecular mechanisms that control these dynamic developmental processes underlying neural circuit assembly, we performed scRNA-seq of PNs from four different developmental stages: 0–6 hr APF, 24–30 hr APF, 48–54 hr APF, and 1–5 days adult (hereafter 0, 24, 48 hr APF and adult) (*Figure 1C*). We used  GH146-GAL4 (*Stocker et al., 1997*) to drive *UAS-mCD8-GFP* (*Lee and Luo, 1999*) expression in most PNs at 24 hr, 48 hr, and adult, which labels ~90 of the estimated 150 PNs in each hemisphere, covering ~40 of the 50 PN types. At 0 hr APF, *GH146-GAL4* also labels cells in the optic lobes (*Figure 1—figure supplement 1A*), which are inseparable from the central brain by dissection. Therefore, we used *VT033006-GAL4* to label PNs at 0 hr

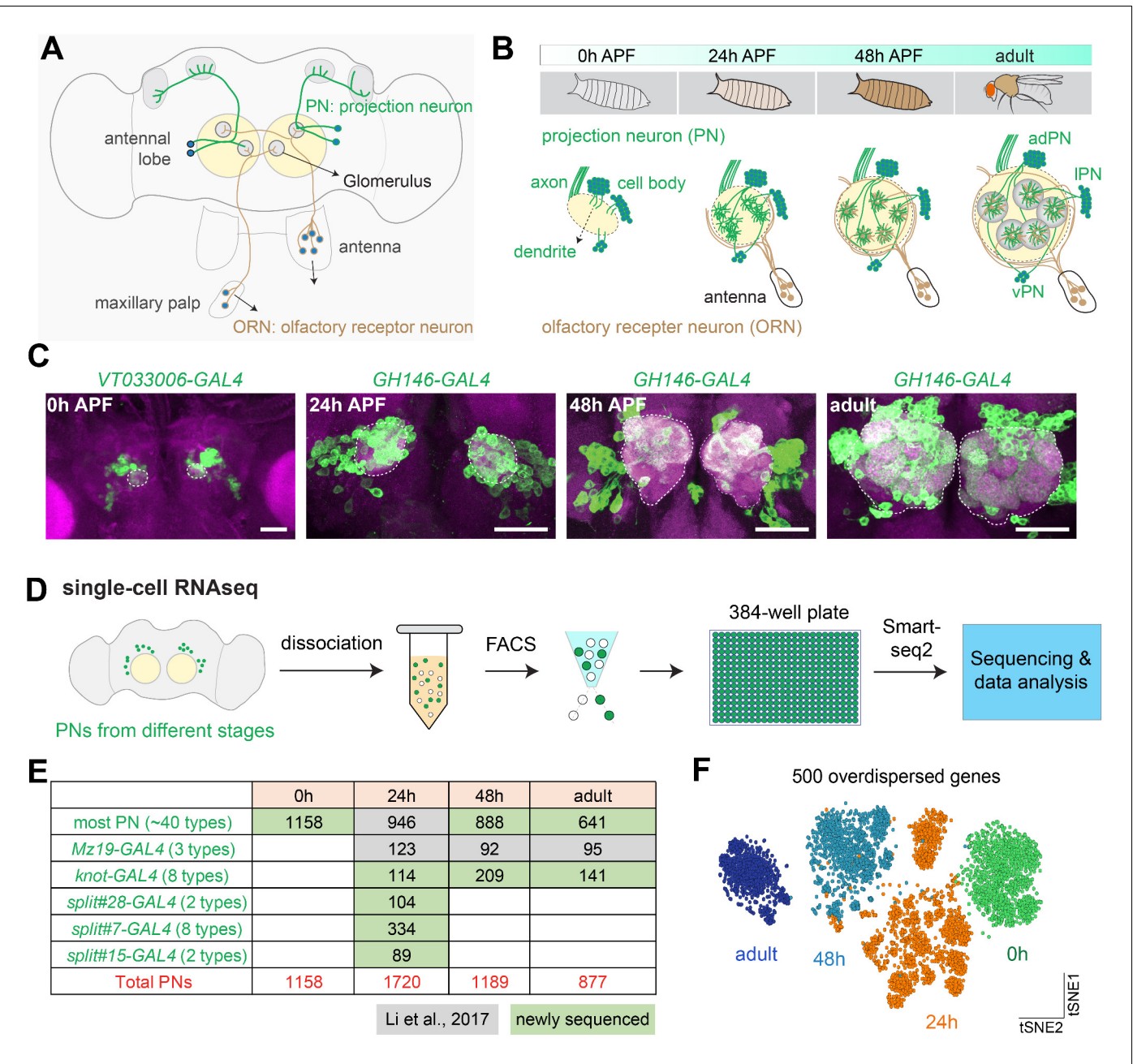

**Figure 1.** Overview of single-cell transcriptomic profiling of *Drosophila* olfactory projection neurons (PNs). (A) Schematic of the adult *Drosophila* olfactory system. Approximately 50 types of olfactory receptor neurons (ORNs) form one-to-one synaptic connections with 50 types of excitatory PNs at 50 glomeruli in the antennal lobe. Illustrated are two types each of ORNs (brown) and PNs (green), as well as two glomeruli to which their axons and dendrites target. (B) Schematic of the developmental process of the adult *Drosophila* olfactory system. The ~50 types of uniglomerular excitatory PNs are from either anterodorsal (adPN) or lateral (lPN) neuroblast lineages. PNs with cell body on the ventral side are inhibitory ventral PNs (vPNs). (C) Representative confocal images of PNs from four different developmental stages, 0 hr APF, 24 hr APF, 48 hr APF, and adult. APF: after puparium formation. Images are shown as maximum z-projections of confocal stacks. Antennal lobes are outlined. Scale bars, 40 μm. (D) Workflow of the single-cell RNA sequencing using plate-based SMART-seq2. FACS: fluorescence-activated cell sorting. (E) Summary of the number of high-quality PNs sequenced at each timepoint and driver lines used. Most PNs refer to PNs sequenced using either *GH146-GAL4* or *VT033006-GAL4*. (F) Visualization of all sequenced PNs from four different developmental stages using tSNE plot. Dimensionality reduction was performed using the top 500 overdispersed genes identified from all sequenced PNs.

The online version of this article includes the following figure supplement(s) for figure 1:

**Figure supplement 1.** Technical characteristics of projection neuron (PN) scRNA-seq.

APF (*Figure 1C* and *Figure 1—figure supplement 1B*; *Tirian and Dickson, 2017*). VT033006-GAL4 labels most PNs from the anterodrosal and lateral lineage, but not PNs from the ventral lineage or anterior paired lateral (APL) neurons like *GH146-GAL4*. It is expressed in ~95 cells that innervate ~44 glomeruli which largely overlap with PNs labeled by *GH146-GAL4* (*Inada et al., 2017*; *Elkahlah et al., 2020*). In addition to PNs labeled by *GH146-GAL4* and *VT033006-GAL4* (we will refer to them as 'most PNs' hereafter), we have collected single-cell transcriptomic data using drivers that only label a small number of PN types for mapping the transcriptomic clusters to anatomically defined PN types.

For scRNA-seq, fly brains with a unique set of PN types labeled using different drivers at each developmental stage were dissected and dissociated into single-cell suspensions. GFP+ cells were sorted into 384-well plates by fluorescence-activated cell sorting (FACS), and sequenced using SMART-seq2 (*Picelli et al., 2014*; *Figure 1D*) to a depth of ~1 million reads per cell (*Figure 1—figure supplement 1C*). On average ~3000 genes were detected per cell (*Figure 1—figure supplement 1D*), and after quality filtering (see Materials and methods), we obtained ~3700 high quality PNs in addition to the previously sequenced ~1200 PNs (*Li et al., 2017*), yielding ~4900 PNs for analysis in this study (*Figure 1E*). All analyzed PNs express high levels of neuronal markers but not glial markers, confirming the specificity of sequenced cells (*Figure 1—figure supplement 1E*). Unbiased clustering using overdispersed genes from all PNs readily separates them into different groups according to their stage (*Figure 1F*), suggesting that gene expression changes across these four developmental stages represent a principal difference in their single-cell transcriptomes.

## Decoding the glomerular identity of transcriptomic clusters by sequencing subsets of PNs at 24 hr APF

PNs labeled by *GH146-GAL4* at 24 hr APF form ~30 distinct transcriptomic clusters. We previously matched six of these transcriptomic clusters to specific anatomically and functionally defined PN types (*Li et al., 2017*), hereafter referred to as 'decoding transcriptomic identity'. Unlike ORNs, whose identities can be decoded using uniquely expressed olfactory receptors (*Li et al., 2020a*), PNs lack known type-specific markers. Instead, PN types are mostly specified by combinatorial expression of several genes (*Li et al., 2017*), making it more challenging to decode their transcriptomic identities.

To circumvent these challenges and decode the transcriptomic identities of more types of PNs, we took advantage of the extensive driver line collection in *Drosophila* (*Luan et al., 2006*; *Jenett et al., 2012*; *Dionne et al., 2018*). We searched for split-GAL4 lines that only labeled a small proportion of all PNs (Yoshi Aso, unpublished data). Using such drivers, we could sequence a few types of PNs at a time, plot those cells with most PNs, and then use differentially expressed markers among them to decode their identities one-by-one. *split#28* GAL4 labeled two types of PNs—those that project their dendrites to the DC3 and DA4l glomeruli in developing and adult animals (*Figure 2A,B*; note that PN types are named after the glomeruli they project their dendrites to). We sequenced those PNs (*split#28+* PNs hereafter) at 24 hr APF. We chose this stage because this is when different PN types exhibit the highest transcriptome diversity as hinted by the number of clusters seen in *Figure 1F* (see following sections for more detailed analysis). To visualize sequenced *split#28+* PNs, we performed dimensionality reduction using 561 genes identified from most 24 hr PNs using Iterative Clustering for Identifying Markers (ICIM), an unsupervised machine learning algorithm (*Li et al., 2017*), followed by embedding in the tSNE space. *Split#28+* PNs (orange dots) fell into two distinct clusters and intermingled with *GH146+* PNs (gray dots) (*Figure 2C*). One cluster mapped to previously decoded DC3 PNs (*Li et al., 2017*), and the other cluster expressed *zfh2* (*Figure 2—figure supplement 1A*). We validated that this cluster indeed represents DA4l PNs by visualizing the expression of *zfh2* in PNs utilizing an intersectional strategy by combining *zfh2-GAL4*, *GH146-Flp*, and *UAS-FRT-STOP-FRT-mCD8-GFP* (hereafter referred to as 'intersecting with *GH146-Flp*') (*Figure 2—figure supplement 1B*).

*split#7* GAL4 labeled three types of PNs in the adult stage (*Figure 2—figure supplement 2A*). However, when we sequenced cells labeled by this GAL4 line at 24 hr APF and visualized them using tSNE, we found eight distinct clusters (*Figure 2F*). We reasoned that this could be due to loss of driver expression in adult stage for some PN types. To test this hypothesis and reveal PNs that are labeled by this driver transiently during development, we used a permanent labeling strategy to label all cells that express *split#7* GAL4 at any time of development (*split#7+* PNs hereafter) by

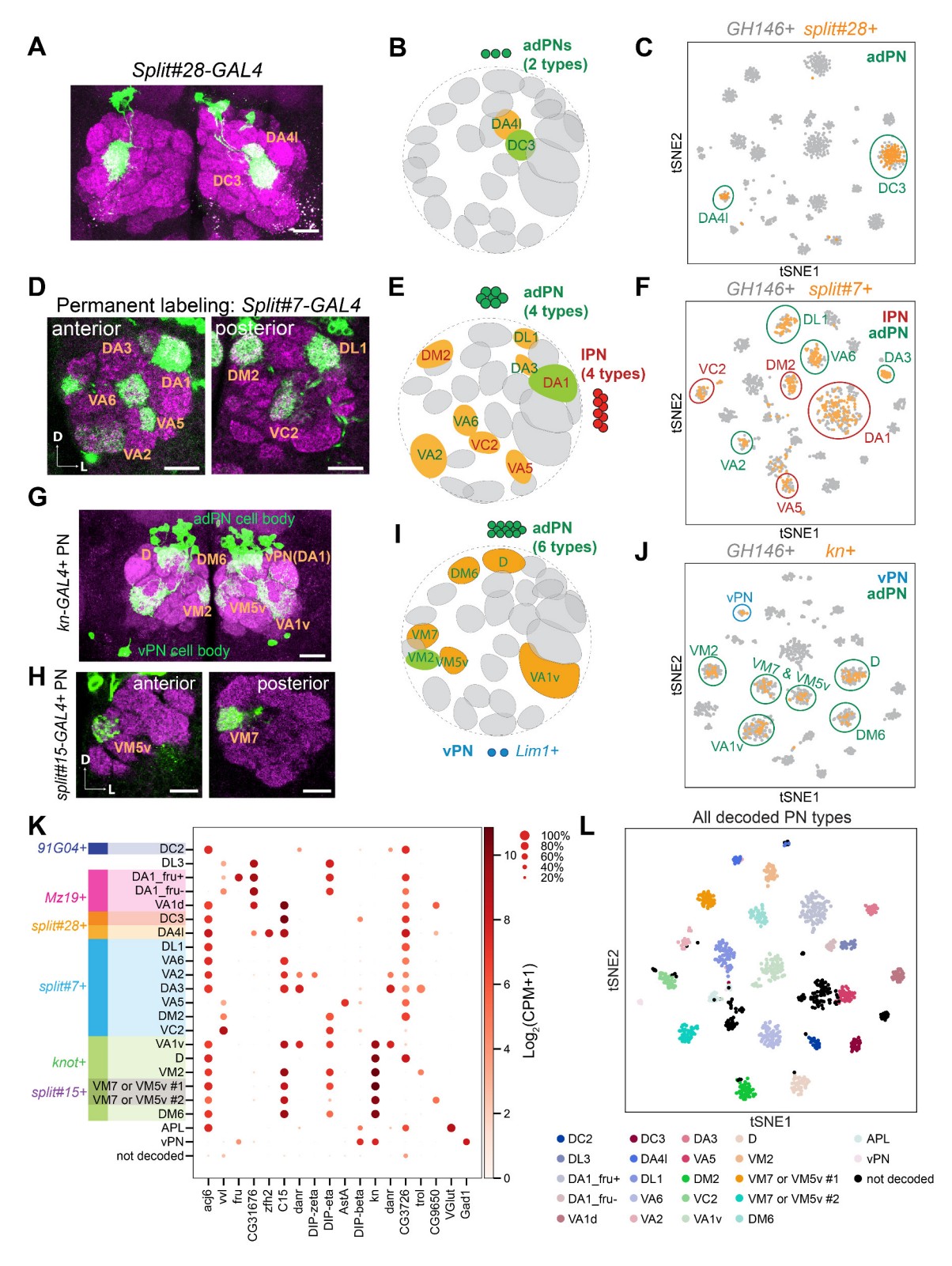

**Figure 2.** Matching 15 transcriptomic clusters to specific projection neuron (PN) types at 24 hr APF. (**A**) Representative maximum z-projection of confocal stacks of *split#28* GAL4 in adults. Dendrites of *split#28* GAL4+ PNs target the DC3 and DA4l glomeruli. (**B**) Diagram of *split#28-GAL4+* PNs. (**C**) tSNE plot showing newly sequenced *split#28-GAL4+* PNs, which form two clusters that can be assigned to DC3 and DA4l PNs (see also ***Figure 2— figure supplement 1***). (**D**) Representative confocal images of *split#7* GAL4 labeled PNs using permanent labeling strategy. One anterior section and

*Figure 2 continued on next page*

*Figure 2 continued*

one posterior section of the antennal lobe are shown. Using permanent labeling, we found that this driver is expressed in eight PN types. Genotype: *split#7-GAL4, UAS-Flp, Actin promoter-FRT-STOP-FRT-GAL4, UAS-mCD8-GFP*. (E) Diagram of *split#7-GAL4+* PNs. *split#7* GAL4 labels eight types of PNs. four from the adPN lineage (green letters) and four from the lPN lineage (red letters). (F) tSNE plot of *split#7* GAL4 PNs with *GH146+* PNs (see *Figure 2—figure supplement 2* for details on the decoding procedure). (G) Representative maximum z-projection of confocal stacks of *kn+* PNs in the adult. *kn-GAL4* was intersected with *GH146-Flp* to restrict the expression of GAL4 in only PNs. (H) Representative confocal images of *split#15* GAL4 in adults, which labels two *kn+* PN types. (I) Diagram showing that *kn+* PNs include six types of adPNs and two vPNs. (J) tSNE plot of *kn-GAL4* PNs with *GH146+* PNs (see *Figure 2—figure supplement 3* for details on the decoding procedure). (K) Dot plot summarizing drivers and marker genes we used to map 21 transcriptomic clusters to 20 PN types [14 adPNs, 5 lPNs—DA1 PNs form two clusters, one *fru+* and one *fru–* (*Li et al., 2017*)—and 1 vPNs] and the anterior paired lateral (APL) neurons at 24 hr APF. Gene expression level [log$_2$(CPM+1)] is shown by the dot color, and percentages of cells expressing a marker are shown by dot size. (L) tSNE plot showing 24 hr APF PNs colored by PN types (*GH146+* PNs with *split#7+/ split#28* PNs to increase cell number in some less abundant PN types). Scale bars, 20 µm. Axes, D (dorsal), L (lateral). In panel B, E, and I, orange glomeruli represent PN types of unknown transcriptomic identity prior to this study. Green glomeruli represent PN types whose transcriptomic identity were previously decoded. Note that the positions of cells on a tSNE plot are dependent on the random initialization of the program as well as every cell present in the dataset, therefore the position of *GH146+* PNs clusters are different when we plot them with different set of newly sequenced PNs (gray in panels C, F, and J).

The online version of this article includes the following figure supplement(s) for figure 2:

**Figure supplement 1.** Validation of DA4l projection neuron (PN) identity.
**Figure supplement 2.** Decoding the identity of *split#7+* projection neurons (PNs).
**Figure supplement 3.** Decoding the identity of *kn+* projection neurons (PNs).

combining it with *UAS-mCD8-GFP, Actin promoter-FRT-STOP-FRT-GAL4*, and *UAS-Flp*. Using this strategy, we observed labeling of eight types of PNs (*Figure 2D*), consistent with number of clusters we observed by sequencing. Among *split#7+* PNs, four types belong to the adPN lineage (*acj6+*) and the other four types belong to the lPN lineage (*vvl+*) (*Figure 2E*). Only one lPN type, DA1 (*CG31676+*), has previously been decoded (*Figure 2—figure supplement 2B*). We identified differentially expressed genes among *split#7+* PNs and obtained existing GAL4 lines mimicking their expression. By intersecting those GAL4 lines with *GH146-Flp*, we mapped all seven previously unknown transcriptomic clusters to seven PN types (*Figure 2—figure supplement 2C-H*; see legends for detailed description).

In addition to screening through collections of existing driver lines, we also utilized scRNA-seq data to find drivers that label a subpopulation of PNs. One such marker was the gene *knot (kn)*, which was expressed in seven transcriptomic clusters among all *GH146+* PNs (*Figure 2—figure supplement 3A*). One of the *kn+* clusters expressing *trol* has been previously mapped to VM2 PNs (*Li et al., 2017*). When *kn-GAL4* was intersected with *GH146-Flp*, six types of adPNs (*acj6+*) and several vPNs (*Lim1+*) were labeled (*Figure 2G,J*). Among the six adPN types, VM7 and VM5v PNs were also labeled by *split#15* GAL4 (*Figure 2H*). Although it has been previously reported that *GH146-GAL4* is not expressed in VM5v PNs (*Yu et al., 2010*), labeling of these PNs when *GH146-Flp* was intersected with either *kn-GAL4* or *split#15* GAL4 indicates that *GH146-Flp* must be expressed in VM5v PNs at some point during development. Using *split#15* GAL4, we were able to decode the two clusters to be either VM7 or VM5v PNs (*Figure 2—figure supplement 3B*). Due to the lack of existing GAL4 drivers for differentially expressed genes between these two clusters, we could not further distinguish them so far; their identities can be decoded by creating new GAL4 drivers in future studies. Other than these two clusters, we were able to match transcriptomic clusters and glomerular types for the rest of *kn+* adPNs one-to-one (*Figure 2—figure supplement 3C–E*). In addition to excitatory PNs, one *kn+* vPN type innervated DA1 glomerulus (because DA1 glomerulus is innervated only by lPNs and vPNs, not adPNs). We found that *DIP-beta* was expressed in one *kn+* vPN cluster but not in lPNs innervating DA1 glomerulus (*Figure 2—figure supplement 3F,G*). Intersecting *DIP-beta-GAL4* with *GH146-Flp* confirmed that *DIP-beta+* vPN indeed targeted their dendrites to DA1 glomerulus, illustrating the *DIP-beta+* vPN cluster to be DA1 vPNs (*Figure 2—figure supplement 3H*).

In summary, by sequencing a small number of known PN types at a time and analyzing the expression pattern of differentially expressed genes, we have now mapped a total of 21 transcriptomic clusters corresponding to anatomically defined PN types at 24 hr APF (*Figure 2K,L*). Ultimately, we aimed to match the transcriptomes of the same PN types across development. As an

intermediate step, we carried out global analysis of gene expression changes across development, which could help us reliably identify transcriptomic clusters representing different PN types at different developmental stages.

## Global gene expression dynamics across four developmental stages

All sequenced PNs segregated into different clusters according to their developmental stages using unbiased, over-dispersed genes for clustering regardless of PN types (*Figure 1F*). Even when we used the genes identified by ICIM for clustering, which emphasizes the differences between different PN types (*Li et al., 2017*), we still observed that individual PNs were separated principally by developmental stages (*Figure 3A*). Together, these observations illustrate global transcriptome changes of PNs from pupa to adult.

To understand what types of genes drive this separation, we searched for genes that were differentially expressed in different developmental stages (*Figure 3B,C*). We clustered the genes into different groups based on their expression pattern throughout development. Seven groups of genes showed clear developmental trends—five groups were down-regulated from pupa to adult and two groups were upregulated (*Figure 3D,E*). Consistent with our previous knowledge, neural development-related genes, including those with functions in morphogenesis and cytoskeleton organization, were enriched in developing PNs (*Figure 3C, D*); genes related to synaptic transmission, ion transport, and behavior, on the other hand, were upregulated in mature PNs (*Figure 3C, E*; *Li et al., 2017*; *Li et al., 2020b*).

## Single-cell transcriptomes of PNs reveal dominant biological processes at different stages of development

Because PN transcriptomes exhibited global development-dependent dynamics, we needed to find a method to reliably and consistently classify transcriptomic clusters representing different PN types at all stages. We first identified informative genes for clustering from each stage using ICIM and used them for further dimensionality reduction. However, using this method, we obtained different numbers of clusters at each stage (*Figure 4A*). Closer examination of each stage revealed unique biological features of PN development.

At 0 hr APF, PNs always formed two distinct clusters—a larger cluster consisting of both adPNs and lPNs, and a smaller one with only adPNs (*Figure 4B*, *Figure 4—figure supplement 2A*). As introduced earlier, although all lPNs and many adPNs are born during the larval stage, some adPNs are born during the embryonic stage. We hypothesized that the smaller cluster could represent embryonically born PNs, which undergo axon and dendrite pruning during early metamorphosis (*Marin et al., 2005*). Neurite pruning in *Drosophila* depends on cell autonomous action (*Lee et al., 2000*) of the steroid hormone ecdysone receptor (EcR) (*Levine et al., 1995*; *Thummel, 1996*; *Schubiger et al., 1998*; *Lee et al., 2000*). Upon binding of the steroid hormone ecdysone, EcR and its co-receptor Ultraspiracle (Usp) form a complex to activate a series of downstream targets, including a transcription factor called Sox14, which in turn promotes expression of the cytoskeletal regulator Mical and Cullin1 SCF E3 ligase (*Figure 4C*; *Lee et al., 2000*; *Kirilly et al., 2009*; *Kirilly et al., 2011*; *Wong et al., 2013*). To test our hypothesis, we examined the expression of genes which are known to participate in neurite pruning and genes that showed elevated expression in the mushroom body γ neurons during pruning (*Alyagor et al., 2018*). We found that *Sox14*, *Mical*, *Cullin1*, and two sorting complexes required for transport (ESCRT) genes—*shrb* and *Vps20*, indeed showed higher expression levels in the smaller cluster (*Figure 4D*). We also confirmed our hypothesis by mapping two types of embryonically born PNs, DA4l and VA6 PNs, to this smaller cluster (*Figure 4—figure supplement 2B*; see mapping details in Figure 7).

At 24 hr APF, we observed the highest number of clusters reflecting different PN types. Moreover, dimensionality reduction using the top 2000 overdispersed genes also showed more distinct clusters at this timepoint compared to the others (*Figure 4—figure supplement 1*). Quantifications of transcriptomic similarity among PNs at each stage indeed confirmed the highest diversity among PNs at 24 hr APF (*Figure 4E–G*). This is likely explained by the fact that at this stage, PNs refine their dendrites to specific regions and begin to prepare themselves as targets for their partner ORN axons. In addition, PN axons at the lateral horn begin to establish their characteristic branching patterns (*Jefferis et al., 2004*). All these processes require high level of molecular diversity among

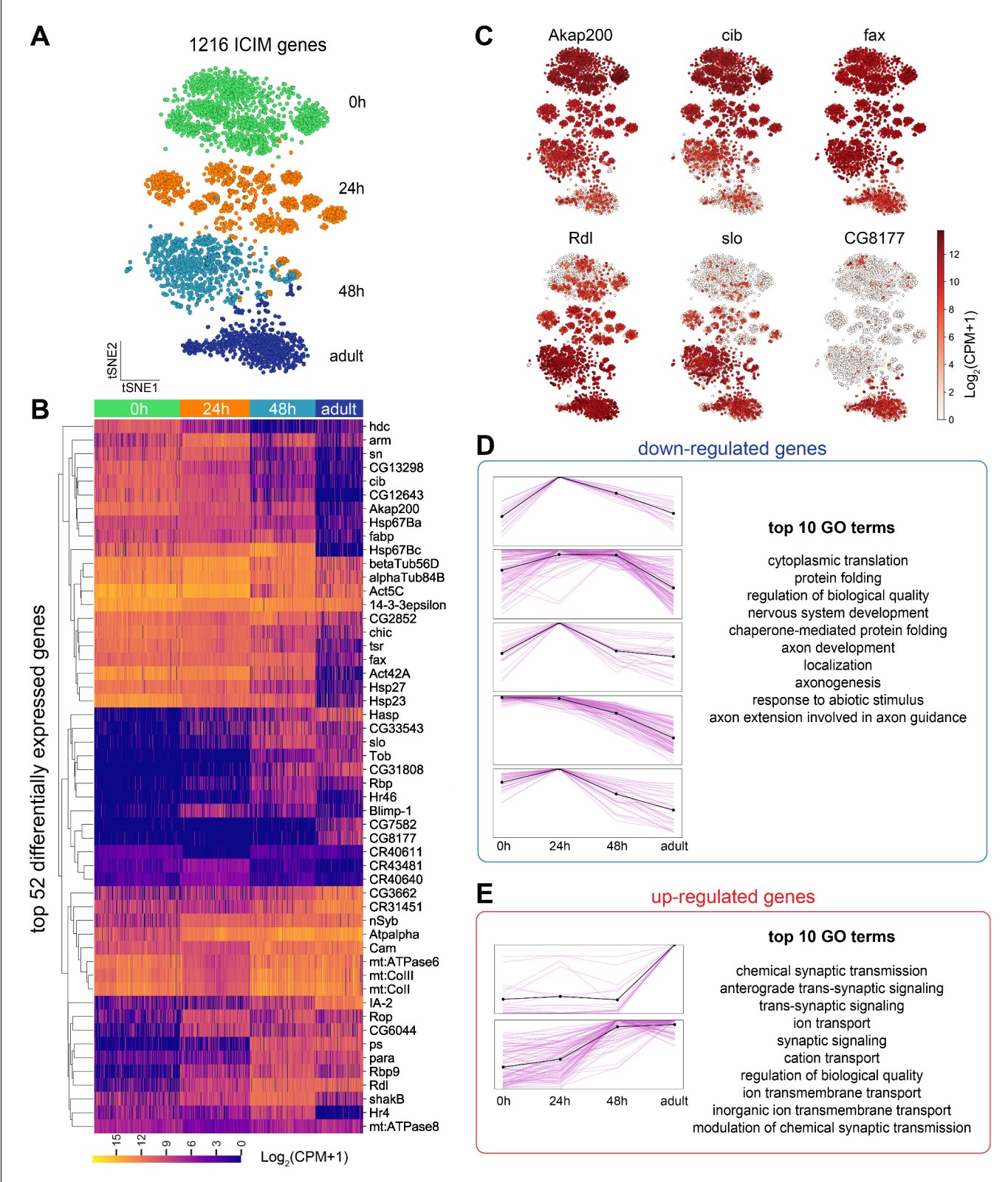

**Figure 3.** Global gene expression dynamics of projection neurons (PNs). (**A**) Visualization of PNs from four different developmental stages: 0 hr APF, 24 hr APF, 48 hr APF, and adult sequenced using either *VT033006-GAL4* or *GH146-GAL4*. tSNE dimensionality reduction was performed using 1216 genes identified by iterative clustering for identifying markers (ICIM) among them. (**B**) Hierarchical heatmap showing the expression of the top 52 out of 474 differentially expressed genes identified among PNs of different developmental stages. (**C**) Examples of the expression of the dynamic genes. Cells are

*Figure 3 continued on next page*

*Figure 3 continued*

colored according to the expression level of each gene. *Akap200* (A kinase anchor protein 200, encodes a scaffolding protein that contributes to the maintenance and regulation of cytoskeletal structure), *cib* (ciboulot, encodes an actin binding protein), and *fax* (failed axon connections, a gene involved in axon development) have the highest expression in early pupal stage and are downregulated gradually. *Rdl* (Resistant to dieldrin, encodes a chloride channel), *slo* (slowpoke, encodes a subunit of calcium-activated potassium channel), and *CG8177* (Anion exchanger 2), are upregulated as PNs develop. (D, E) Top 474 differentially expressed genes can be divided into eight groups based on their dynamic profiles—two groups without obvious developmental trend (not shown), five groups of down-regulated genes (D), and two groups of upregulated genes (E). Pink lines represent individual genes and the black line shows mean expression of genes in each group. The highest expression is normalized as one for all genes. The top 10 Gene Ontology (GO) terms for upregulated and downregulated genes are shown on right.

different PN types to ensure precise wiring, warranting more distinction between their transcriptomes at this stage.

In contrast to the high transcriptomic diversity in 24 hr APF PNs, adult PNs only formed three clusters (*Figure 4A* bottom, indicated by dashed lines). The three clusters represent cholinergic excitatory PNs (marked by *VAChT*), and two *Gad1+* GABAergic inhibitory cell types—vPNs and APL neurons (*VGlut+*), respectively (*Figure 4H*). This is likely because after wiring specificity is achieved, all excitatory PNs may perform similar functions, but distinct from inhibitory neuronal types.

Thus, at different developmental stages, the differentially expressed genes we identified all revealed the most defining biological processes those neurons are undertaking. Our observations showed that PN transcriptomes reflect the pruning process of embryonically born PNs at 0 hr APF, PN type and wiring distinction at 24 hr APF, and neurotransmitter type in adults.

## Identifying PN types at all developmental stages

With the exception of the 24 hr APF PNs, gene sets identified from each of the other stages could not resolve distinct clusters reflecting PN type diversity (*Figure 4*). Therefore, we tried to use the genes identified by ICIM from 24 hr APF PNs to cluster PNs of the other stages. We found that this gene set outperformed all other gene sets in separating different PN types at all timepoints (*Figure 5A*). In fact, most gene sets found by different methods at 24 hr APF, including overdispersed genes, ICIM genes, as well as differentially expressed genes between different clusters, exceeded gene sets identified at other stages for clustering PNs according to their types (data not shown), further confirming that transcriptomes of 24 hr APF PNs carry the most information for distinguishing different PN types, even for other developmental stages.

Following this observation, we decided to use differentially expressed genes between 24 hr PN clusters for PN-type identification for all stages. We applied *meta*-learned *r*epresentations for *s*ingle-cell data (MARS) for identifying and annotating cell types (*Brbić et al., 2020*). MARS learns to project cells using deep neural networks in the latent low-dimensional space in which cells group according to their cell types. We used 24 hr APF, the stage with highest transcriptome diversity, as the starting annotated dataset to learn shared low-dimensional space for 48 hr APF, 0 hr APF, and eventually the adult dataset. Using this approach, we found ~30 cell types in each stage (*Figure 5B*). Independently, we also validated MARS cluster annotations using two distinct methods: HDBSCAN clustering based on tSNEs and Leiden clustering based on neighborhood graphs (*Figure 5—figure supplement 1*; *Blondel et al., 2008*; *Levine et al., 2015*; *Traag et al., 2019*). Clusters identified by HDBSCAN and Leiden largely agreed with MARS annotations, confirming the reliability of MARS. We compared cluster annotations by these three methods to known PN types at 24 hr APF (*Figure 5—figure supplement 1C*) and found that even at this stage, MARS performed better at segregating some closely related clusters representing multiple PN types (*Figure 5—figure supplement 1D*). At 0 hr APF and the adult stage, MARS identified more clusters compared to the other methods, demonstrating the robustness of MARS at identifying unique cell types.

## Matching the same PN types across four developmental stages

We next sought to match transcriptomes of the same PN type across different developmental stages. We first tried to apply some batch correction methods, including Harmony, BBKNN, ComBat, and Scanorama, to our dataset to correct for the transcriptomic changes of PNs throughout development (*Hie et al., 2019*; *Johnson et al., 2007*; *Korsunsky et al., 2019*; *Polański et al., 2020*). For all batch correction methods attempted, we observed instances of (1) PNs of the same

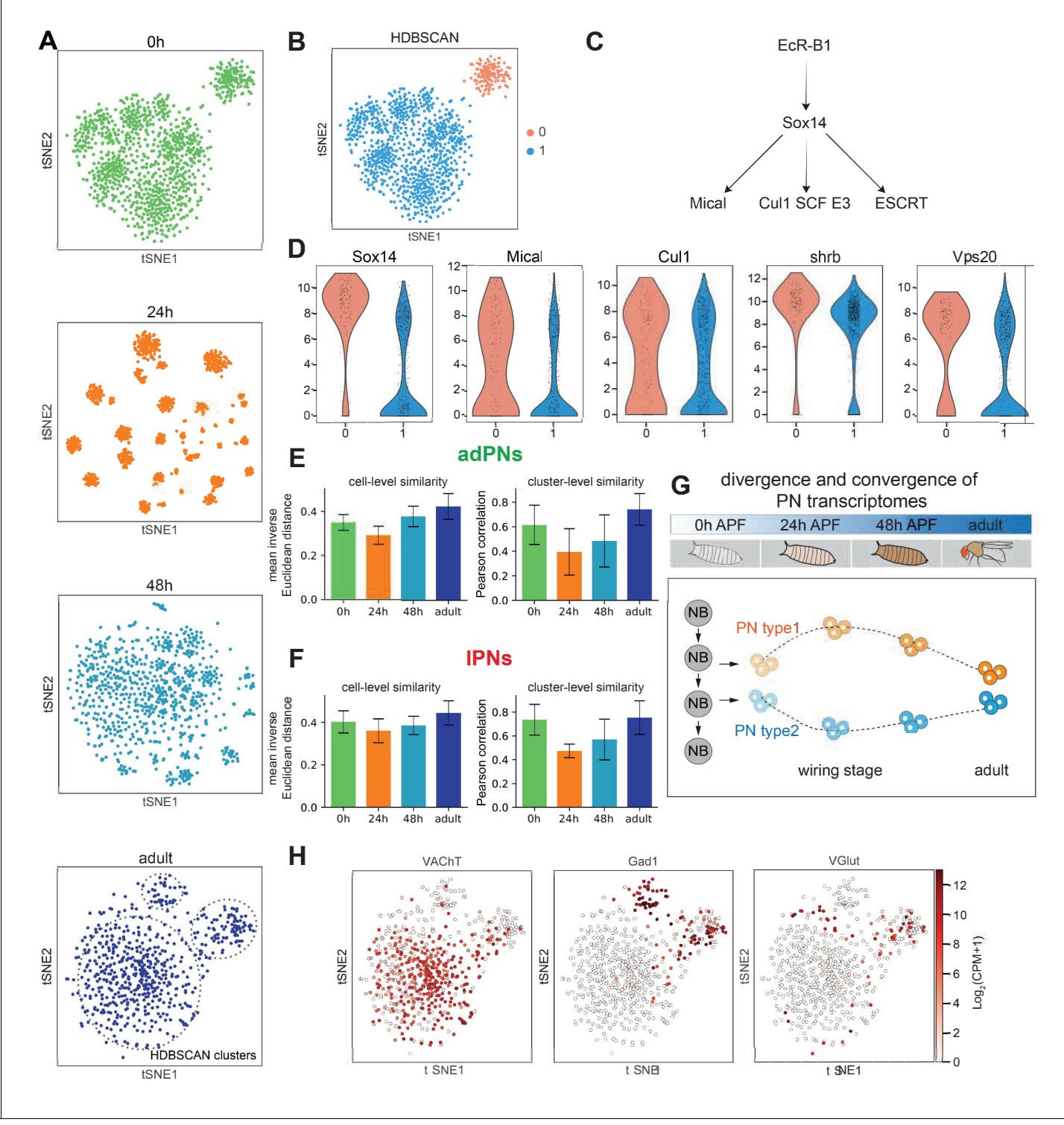

**Figure 4.** PN transcriptomes show distinct features at different stages of development. (**A**) Visualization of most projection neurons (PNs) from 0 hr APF, 24 hr APF, 48 hr APF, and adults using tSNE based on genes identified by ICIM at each stage. Adult clusters (circled) are identified using HDBSCAN. (**B**) Clustering of 0 hr APF PNs using HDBSCAN identified two clusters. (**C**) Part of the molecular pathways critical for neurite pruning in *Drosophila*. (**D**) Genes whose function have been implicated in neurite pruning have higher expression in cluster 0: *Sox14* (p-value: 5.01E-51), *Mical* (p-value: 1.49E-09), *Cul1* (p-value: 8.15E-4), *shrb* (p-value: 6.37E-19), and *Vps20* (p-value: 1.23E-17) ( Mann-Whitney U test). (**E, F**). PN transcriptomic similarity calculated at the cell level (mean inverse Euclidean distance calculated using 1216 ICIM genes identified from PNs of all four stages) and the cluster level (Pearson correlation calculated using differentially expressed genes identified from 24 hr PN clusters) for adPNs (**E**) (0 hr APF: 587 cells, cell-

*Figure 4 continued on next page*

*Figure 4 continued*

level similarity [mean ± standard deviation]: 0.350 ± 0.036, 15 clusters, cluster-level similarity [mean ± standard deviation]: 0.615 ± 0.160; 24 hr APF: 547 cells, cell-level similarity: 0.292 ± 0.041, 15 clusters, cluster-level similarity: 0.395 ± 0.189; 48 hr APF: 301 cells, cell-level similarity: 0.377 ± 0.046, 13 clusters, cluster-level similarity: 0.484 ± 0.212; adult stage: 209 cells, cell-level similarity: 0.422 ± 0.058, 15 clusters, cluster-level similarity: 0.741 ± 0.129) and lPNs (F) (0 hr APF: 484 cells, cell-level similarity: 0.402 ± 0.052, 10 clusters, cluster-level similarity: 0.736 ± 0.129; 24 hr APF: 354 cells, cell-level similarity: 0.360 ± 0.056, 10 clusters, cluster-level similarity: 0.474 ± 0.057; 48 hr APF: 296 cells, cell-level similarity: 0.385 ± 0.043, 10 clusters, cluster-level similarity: 0.570 ± 0.171; adult stage: 191 cells, cell-level similarity: 0.444 ± 0.057, eight clusters, cluster-level similarity: 0.754 ± 0.141). (G) Schematic summary of PN transcriptome similarity changes from early pupal stage to adulthood. PN diversity peaks during circuit assembly around 24 hr APF and gradually diminishes as they develop into mature neurons. (H) Expression of *VAChT*, *Gad1*, and *VGlut* in adult PNs.

The online version of this article includes the following figure supplement(s) for figure 4:

**Figure supplement 1.** Visualization of most projection neurons (PNs) at different stages using tSNE.

**Figure supplement 2.** Embryonically born and larval born projection neurons (PNs) at 0 hr APF.

type at the same stage split into different clusters; (2) PNs of different types merge into the same cluster; (3) no distinguishable cluster formation for many PNs in stages other than 24 hr APF. Therefore, we needed to develop alternative approaches to reliably match transcriptomes of same PN types across different developmental stages. To perform this task, we first used *kn+* PNs as test case. We collected PNs labeled by *kn-GAL4* from 24 hr APF, 48 hr APF, and adult brains for scRNA-seq (*Figure 6A*). Dimensionality reduction of these cells showed a consistent number of clusters across stages (*Figure 6B*). One exception is an extra vPN cluster observed at 48 hr APF and adult stages. This discrepancy with 24 hr APF data is likely caused by the lower number of vPNs sequenced at 24 hr APF.

When *kn+* PNs from all three stages were plotted together, all adPNs (*acj6+* clusters on the upper side) formed relatively distinct clusters and did not intermingle with adPNs from the other

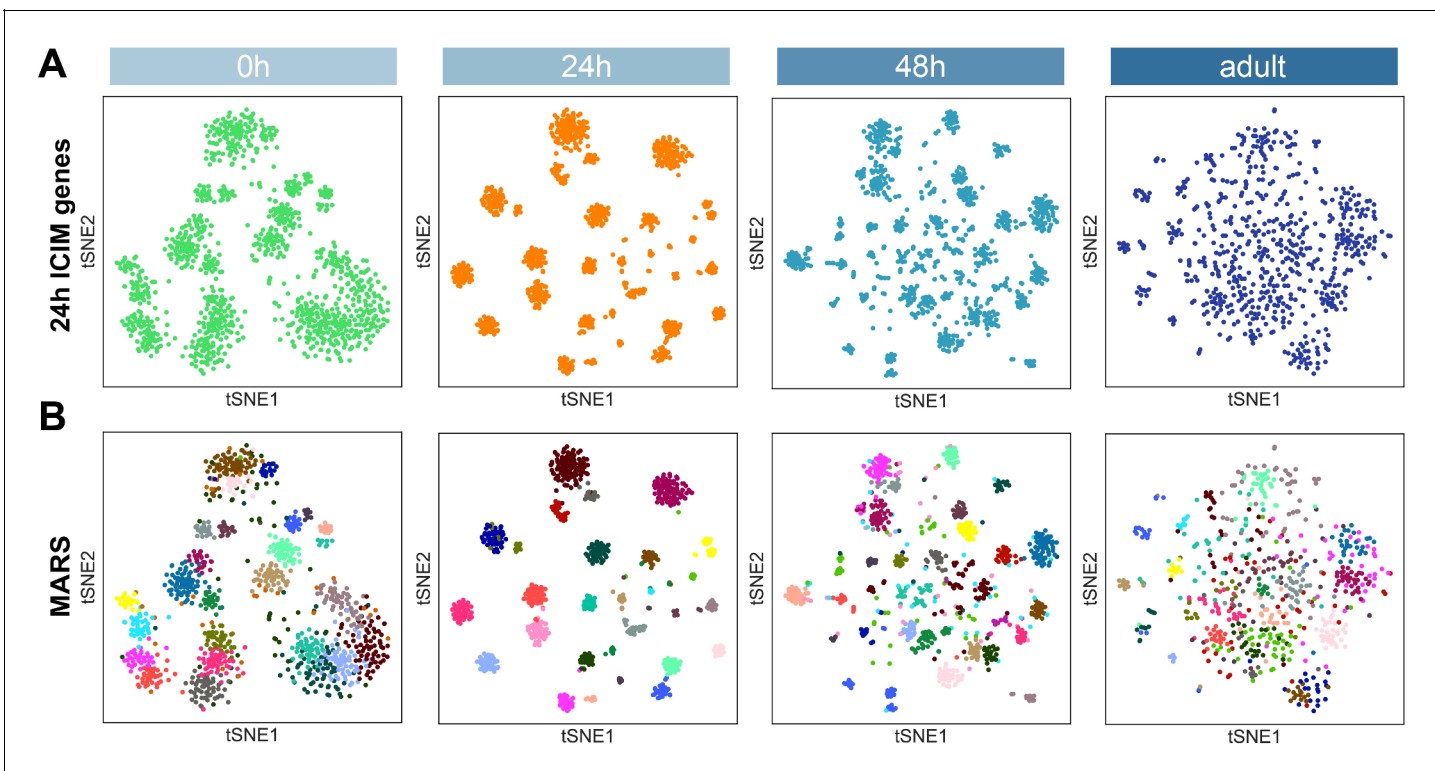

**Figure 5.** Projection neuron (PN) type identification by MARS. (A) Dimensionality reduction of most PNs at four developmental stages by 561 ICIM genes found at 24 hr APF. (B) PN types identified by MARS. Different MARS clusters are illustrated in different colors.

The online version of this article includes the following figure supplement(s) for figure 5:

**Figure supplement 1.** Projection neuron (PN) type identification using two other independent methods.

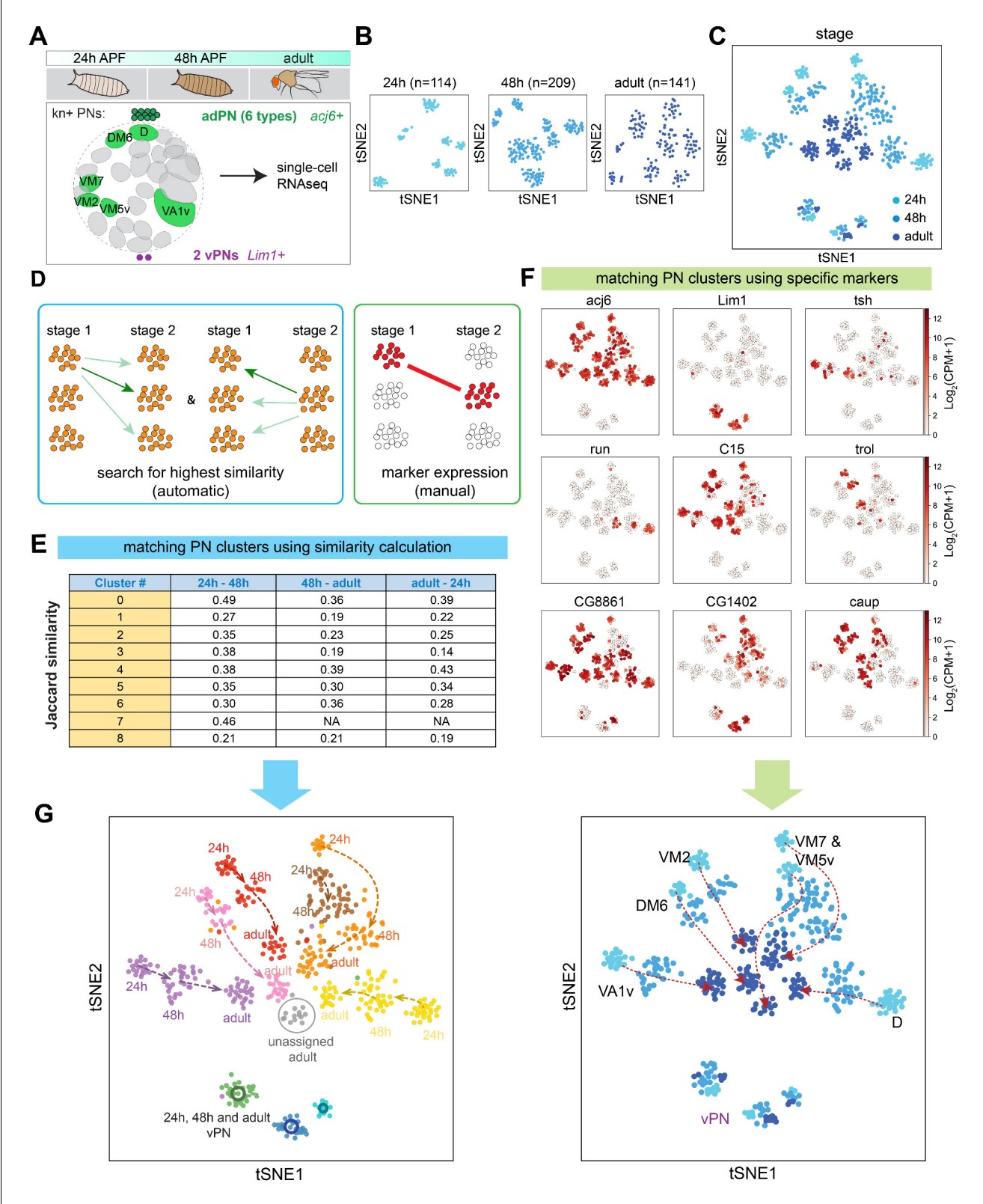

**Figure 6.** Two complementary approaches to match transcriptomic clusters representing same projection neuron (PN) types at different developmental stages. (**A**) scRNA-seq was performed for *kn+* PNs from three different developmental stages: 24 hr APF, 48 hr APF, and adult. (**B**) tSNE plots showing *kn+* PNs from three different stages, plotted separately. Cells are clustered according to 24 hr ICIM genes. Cell numbers are indicated. (**C**) *kn+* PNs from three different stages plotted in the same tSNE plot. Cells are clustered according to 24 hr ICIM genes. (**D**) Two approaches were used for

*Figure 6 continued on next page*

*Figure 6 continued*

matching the same PN types at different stages: (1) automatic prediction by calculating the transcriptomic similarity between clusters at two stages (2) manual matching of clusters using specific markers or marker combinations. (E) Jaccard similarity index of automatically matched transcriptomic clusters from different stages. Clusters #7 (brown cells in panel G) in 24 hr and 48 hr APF do not match with any cluster in the adult stage; therefore, the similarity calculation is left as not applicable (NA). (F) Examples of markers used to manually match transcriptomic clusters representing the same PN types across different stages. (G) All *kn+* PN types (six adPNs and three vPNs) are matched from three different stages. Two independent approaches (automatic and manual) produced similar results.

The online version of this article includes the following figure supplement(s) for figure 6:

**Figure supplement 1.** *kn+* adPN transcriptomes become more similar as development proceeds.

timepoints (*Figure 6C*), reflecting substantial changes in the transcriptome of the same type of PNs across development. To match the same type of PNs, we took two independent approaches (*Figure 6D*). In the first approach, clusters were automatically matched based on their transcriptomic similarity. Briefly, we identified a set of genes that were differentially expressed in each cluster compared to all the rest at the same stage. Then, we calculated the percentage of genes shared between each pair of clusters across two stages (Jaccard similarity index) (*Figure 6E*). If two clusters from two stages both had the highest similarity score with each other, we considered them to be matched. In the second approach, we used markers that were expressed in a consistent number of clusters at each stage. Those markers, or marker combinations, were used to manually match the same type of PNs (some example markers used are shown in *Figure 6F*). Using these two approaches, we were able to match the same types of PNs across three developmental stages, and the results from the two approaches consistently agreed with each other (*Figure 6G*). In addition, these data further validated an earlier conclusion (*Figure 4*) that as development proceeds from 24 hr APF and 48 hr APF to adults, the transcriptomic difference between identified PN types becomes smaller (*Figure 6G*; quantified in *Figure 6—figure supplement 1*).

We next applied the same approaches for matching *kn+* PN types across three stages to match most PNs (sequenced using either *GH146-GAL4* or *VT033006-GAL4*) across four stages (*Figure 7A*). In addition to marker gene expression, we also used subset of PNs we had sequenced from different stages to manually match PN types (*Figure 7—figure supplement 1A–D*). For the manually matched PN types with known identity, we summarized markers and marker combinations we used in a dot plot, where both average expression as well as percentage of cells expressing each marker were shown (*Figure 7—figure supplement 2*). Using both manual and automatic approaches, we were able to match many PN types across two or more developmental stages (*Figure 7B*), which includes 18 PN types that we have decoded in *Figure 2* as well as seven transcriptomic clusters with unknown identity. The majority of the PNs we matched were confirmed by both the automatic (transcriptomic similarity-based) and manual (marker-based) methods (*Figure 7C* and *Figure 7—figure supplement 1E*).

## PN types with adjacent birth order share more similar transcriptomes at early stages of development

Previous works have shown that the PN glomerular types are prespecified by the neuroblast lineages and birth order within each lineage (*Jefferis et al., 2001*; *Marin et al., 2005*; *Yu et al., 2010*; *Lin et al., 2012*; *Figure 8A*). Having decoded the transcriptomic identities of different PN types at different timepoints, we can now ask the extent to which transcriptomic similarity is contributed by lineage and birth order, and whether these contributions persist through development.

To address these questions, we performed hierarchical clustering on all excitatory PN clusters we identified from each timepoint. We plotted the dendrogram and the correlation between each pair of clusters (*Figure 8—figure supplement 1*). We observed some lineage-related similarity between PN types at 0 hr APF: transcriptomes of PNs from the same lineage tended to be clustered together in the dendrogram and their correlations are higher, although the relationship was not absolute. Such similarity was gradually lost as development proceeded (as inferred by both the dendrogram as well as correlation between PNs from the same lineage). Interestingly, we noticed that some PNs with adjacent birth order appeared to be neighbors in the dendrogram at 0 hr and 24 hr APF.

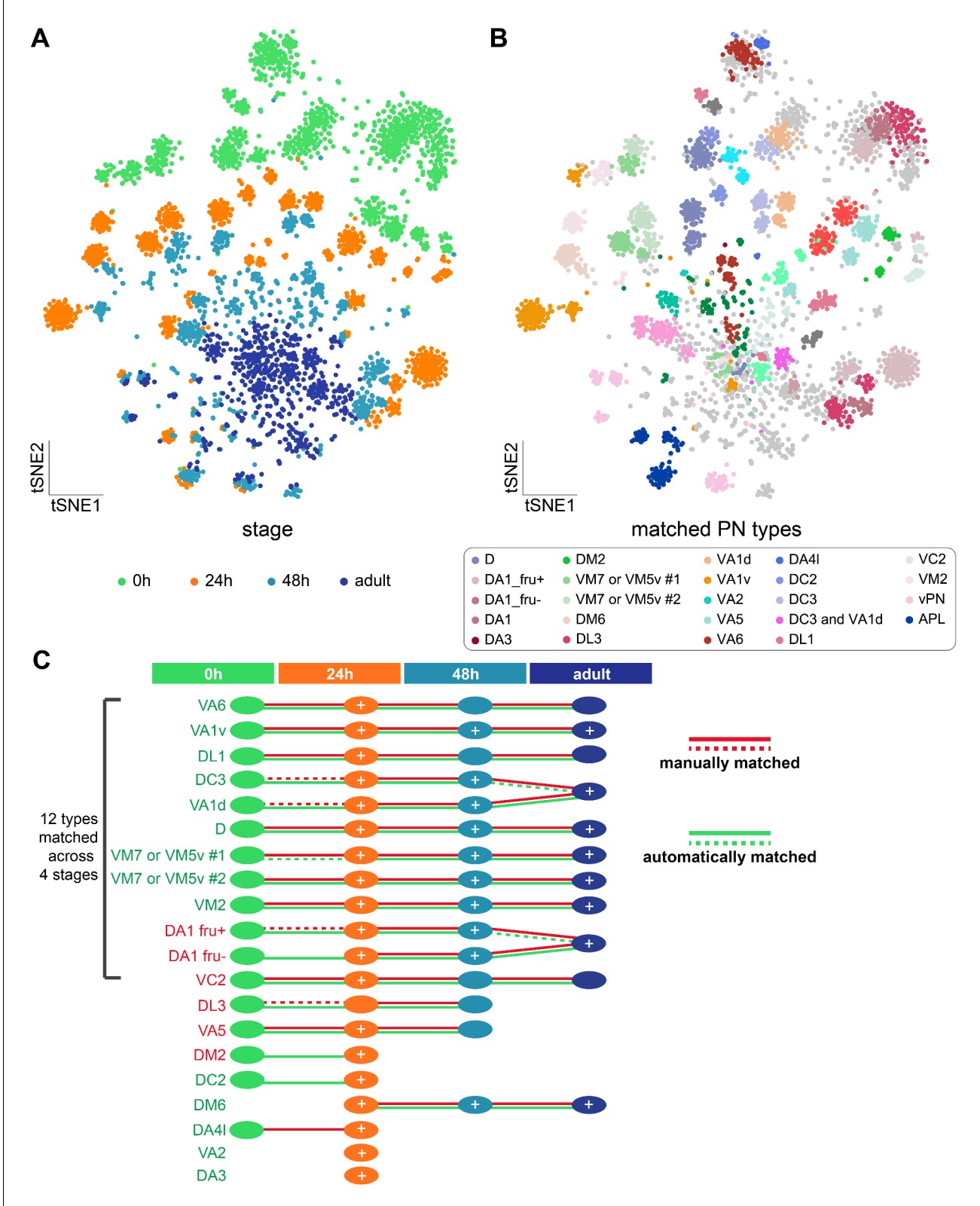

**Figure 7.** Matching transcriptomic cluster representing the same projection neuron (PN) types across four developmental stages. (A) Visualization of most PNs at four different developmental stages: 0 hr APF, 24 hr APF, 48 hr APF, and adult. A total of 561 ICIM genes at 24 hr APF PNs were used for dimensionality reduction. (B) Visualization of the same types of PNs at all developmental stages. Clusters with the same color represent same neuronal type. Light gray dots indicate cells that have neither been decoded nor matched. (C) Summary of transcriptomic clusters mapped to known PN types at

*Figure 7 continued on next page*

*Figure 7 continued*

different developmental stages. Solid red-lines indicate clusters we can unambiguously match using marker combinations; dashed red-lines indicate PN types we can narrow down to less than three transcriptomic clusters. Solid green-lines indicate clusters that are two-way matched automatically (two clusters from two stages are the most similar to each other); dashed green-lines indicates clusters that are one-way matched automatically (one cluster is the most similar with the other but not the other way around). Circles with white '+' indicate PN types that have been sequenced and confirmed at that stage using additional GAL4 lines (see *Figure 7—figure supplement 1*).

The online version of this article includes the following figure supplement(s) for figure 7:

**Figure supplement 1.** Supporting evidence for matching projection neuron (PN) types across developmental stages.

**Figure supplement 2.** Markers used for manually matching projection neurons (PNs).

To further investigate the relationship between birth order of PNs and their transcriptomic similarity, we selected all decoded PNs from the anterodorsal lineage, ordered them according to their birth order, and computed their correlation (*Figure 8B*). 0 hr APF adPNs showed high correlation between their birth order and their transcriptomic similarity, as indicated by the high correlations in boxes just off the diagonal line. To test if the transcriptomic similarity of adPNs indeed covaries with their birth order, we performed permutation tests, comparing the Spearman correlations between birth-order ranking and transcriptomic similarity ranking (*Figure 8C*, see Materials and methods for details). The results confirmed that 0 hr and 24 hr APF PNs, but not 48 hr APF and adult PNs, exhibited high correlations between their birth orders and transcriptomic similarities. In addition, developmental trajectory analysis of adPNs born at the larval stage using Monocle 3 (*Cao et al., 2019*) also showed that the unbiased pseudo time recapitulated their birth order (*Figure 8D*).

A previous study profiled the transcriptomes of PN neuroblasts at various larval stages and identified 63 genes with temporal gradients (*Liu et al., 2015*). Among those genes, the authors have validated that two RNA-binding proteins, Imp and Syp, regulate the fate of PNs born at different times. Therefore, we analyzed expression of these 63 genes at 0 hr APF to see if any of these genes with temporal gradients has persisted expression in postmitotic PNs. We found 15 out of the 63 genes (including *Imp* but not *Syp*) maintained some temporal gradient patterns according to their birth order at 0 hr APF (*Figure 8E*) but not at the later stages (data not shown). This result suggested that the expression of a subset of birth order-related genes in adPN neuroblast, including a cell-fate regulator, is maintained in postmitotic PNs till early pupal stage.

In summary, our data demonstrated that PN types with adjacent birth order shared more similar transcriptomes, reflecting temporal gene expression dynamics of their progenitor. Such transcriptomic similarity was maintained at early pupal stages and was gradually lost as PNs mature.

## Differentially expressed genes in different PN types

Hierarchical clustering on the principal components calculated using the entire gene matrix indicates that the similarities between different PN types are not fixed across development (*Figure 8—figure supplement 1*). This suggests that the differentially expressed genes in PNs differ across developmental stages. Identifying differentially expressed (DE) genes, especially among those that we have matched across multiple developmental stages (*Figure 7*), can allow us to investigate expression dynamics in different PN types and also reveal interesting molecules for future studies.

We consider a gene to be differentially expressed if it has an adjusted p-value of less than 0.01 by Mann-Whitney U test in at least one cluster compared to the rest of the clusters. Using this criterion, we found around 500 DE genes at 24 hr APF, 48 hr APF, and the adult stage (*Figure 9A*). At 0 hr APF, many more DE genes were identified. The larger gene set at this stage is mostly contributed by the embryonically born PNs (1015 out of 1393 genes), which have transcriptomically distinct features because these neurons undergo axon and dendrite pruning (*Figure 4A–D*). We intersected the four lists of DE genes to find genes that are differentially expressed throughout development. This resulted in 103 genes, 52 of which were differentially expressed among the 12 PN types we matched across all four stages. Among the DE genes that are differentially expressed in all four stages, we observed an over-representation of transcription factors (TFs) and cell surface molecules (CSMs) compared to their genome-wide fractions (*Figure 9B*). Previous studies have shown that genes in these two categories play critical roles in PN wiring (*Hong and Luo, 2014*; *Li et al., 2017*). We

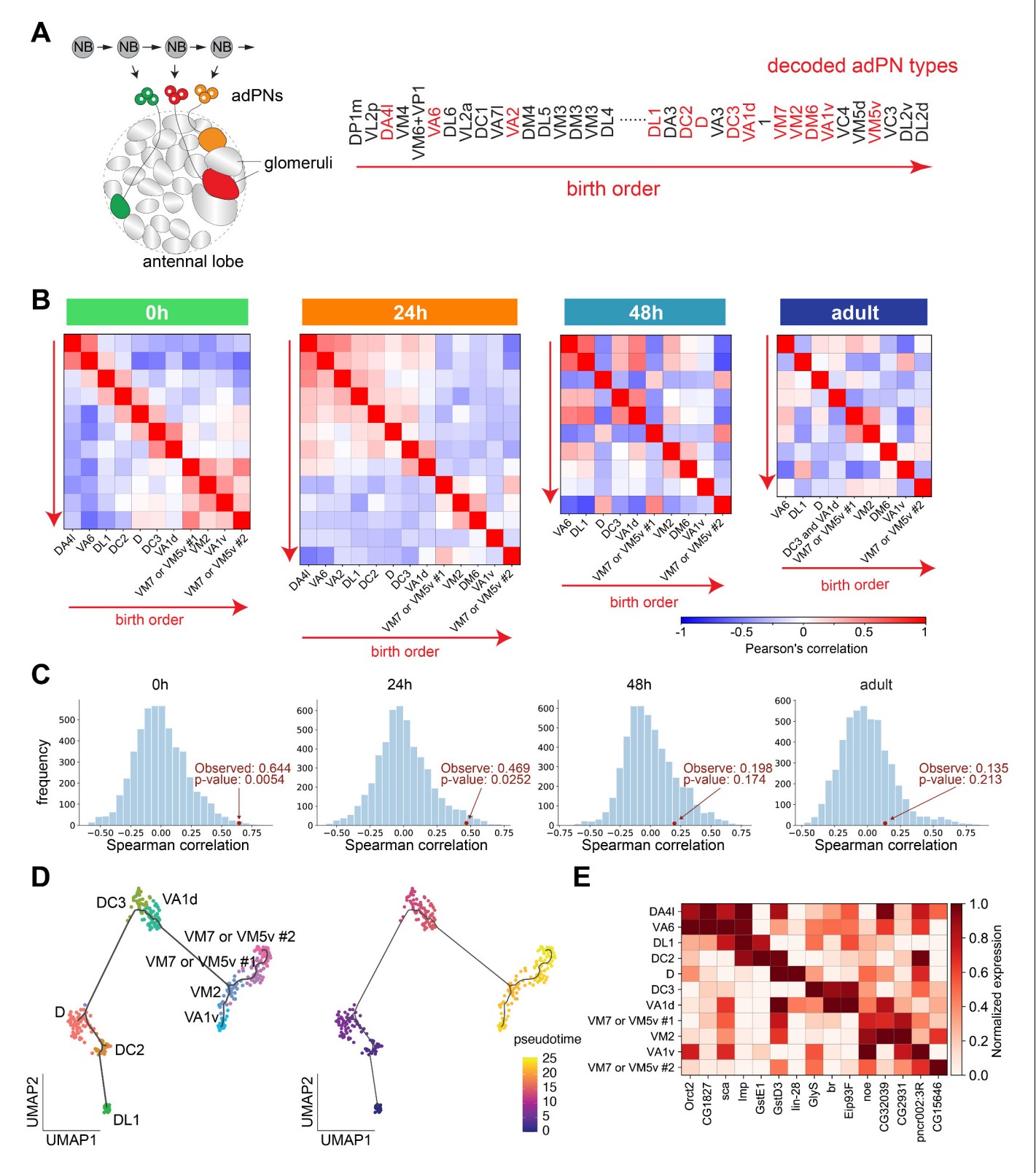

**Figure 8.** Projection neuron (PN) types with adjacent birth order share more similar transcriptomes at early pupal stages. (**A**) Different PN types born from a common neuroblast follow a stereotyped sequence. The birth order of PNs determines to which glomerulus their dendrites target. The birth order of adPNs are shown on right. PN types with known transcriptomic identities at any of the four stages are highlighted in red. (**B**) Correlation matrix of the transcriptomes of adPNs with known identities (Pearson's correlation). PN types are ordered according to their birth order. At 0 hr and 24 hr APF,

*Figure 8 continued on next page*

*Figure 8 continued*

PN types with birth orders adjacent to each other exhibit the highest correlations in their transcriptomes, as indicated by high correlations in boxes just off the diagonal line. (C) Results of permutation test under the null hypothesis that the ranks of adPN transcriptomic similarity do no covary with the ranks of birth order. Observed values is the average Spearman correlation of eight adPN types decoded in all four stages (red dot). The distribution is the average Spearman correlations obtained by randomly permuting the birth order for 5000 iterations (histogram). (D) Developmental trajectory analysis showing an unbiased pseudo time of 0 hr APF adPNs (embryonically born types excluded). The pseudo time roughly matches their birth order. (E) Expression levels of 15 genes in adPNs with known identity at 0 hr APF. These genes have been shown to exhibit temporal expression gradient in PN neuroblasts (*Liu et al., 2015*). The highest expression is normalized as one for all genes.

The online version of this article includes the following figure supplement(s) for figure 8:

**Figure supplement 1.** Hierarchical clustering of all excitatory projection neurons (PNs).

therefore further explored the expression pattern of these genes (*Figure 9C* and *Figure 9—figure supplements 1* and *2*).

While the majority of TFs are expressed in both lineages, expression of a small fraction of TFs is lineage-specific. For example, expression of *acj6*, *kn*, *C15*, and *salr* is limited to PNs from the antero-dorsal lineage, whereas *vvl* and *unpg* are only expressed in PNs from the lateral lineage (*Figure 9C* and *Figure 9—figure supplement 1*). Furthermore, although TFs are generally expressed in a binary fashion throughout development (*Figure 9C* and *Figure 9—figure supplement 1*), many CSMs exhibit graded expression with complex temporal dynamics (*Figure 9D* and *Figure 9—figure supplement 2*). This is consistent with observations made from single-cell transcriptome studies in the developing *Drosophila* optic lobe (*Kurmangaliyev et al., 2020*; *Özel et al., 2021*). Among the CSMs that are differentially expressed in any of the four stages, we observed many molecules in protein families that have been implicated in wiring, including Beaten Path (Beat), Dpr, DIP, Dscam, Fasciclin (Fas), and Robo (*Figure 9—figure supplement 2*; *Kolodkin and Tessier-Lavigne, 2011*; *Sanes and Zipursky, 2020*). Thus, this differentially expressed gene list may contain an enriched set of wiring-related molecules, some of which have been studied in the context of wiring. Therefore, our data can serve as a useful resource for future studies of wiring specificity. On the other hand, we note that some genes with differential expression pattern at the protein level, such as Ten-a and Ten-m (*Hong et al., 2012*), do not exhibit obvious differential expression at the mRNA level. This highlighted the existence of post-transcriptional regulation for some genes that are not captured by transcriptomic analysis.

## Genes involved in metabolism and neuronal signaling are differentially expressed among adult PNs

Our analyses have shown that transcriptomic differences between different PN types diminish as development proceeds (*Figure 4*). However, different PN types in adults still exhibited differential gene expression (*Figure 9*). Such differential expression could be contributed by residual developmentally differentially expressed genes, by new categories of differentially expressed genes in mature PNs reflecting functional differences between different PN types, or a combination of both. To distinguish between these possibilities, we compared DE genes among different transcriptomic clusters of PNs at 24 hr APF and at the adult stage.

We found that more than a third of the DE genes were shared between these two stages (*Figure 10A*). Gene Ontology analysis revealed that these shared genes were predominately related to neural development (*Figure 10B*, middle). These data suggested that some DE genes found among adult PN types were developmentally differentially expressed genes, some of which could play a role in the maintenance of adult nervous system structures.

Interestingly, many Gene Ontology terms related to the physiological properties of PNs were observed among the adult-only DE genes (*Figure 10B*, bottom). In addition, we observed several ion-channels and neurotransmitter receptors in the list of CSMs with differential expression pattern (*Figure 9—figure supplement 2*). Indeed, several adult DE genes belong to the ion channels or transmembrane receptor (including neurotransmitter receptors and G-protein-coupled receptors) gene groups (*Figure 10C*). These results demonstrated that PN types in adults acquire new categories of differentially expressed genes, and those genes might lead to differences in the physiological properties between different PN types.

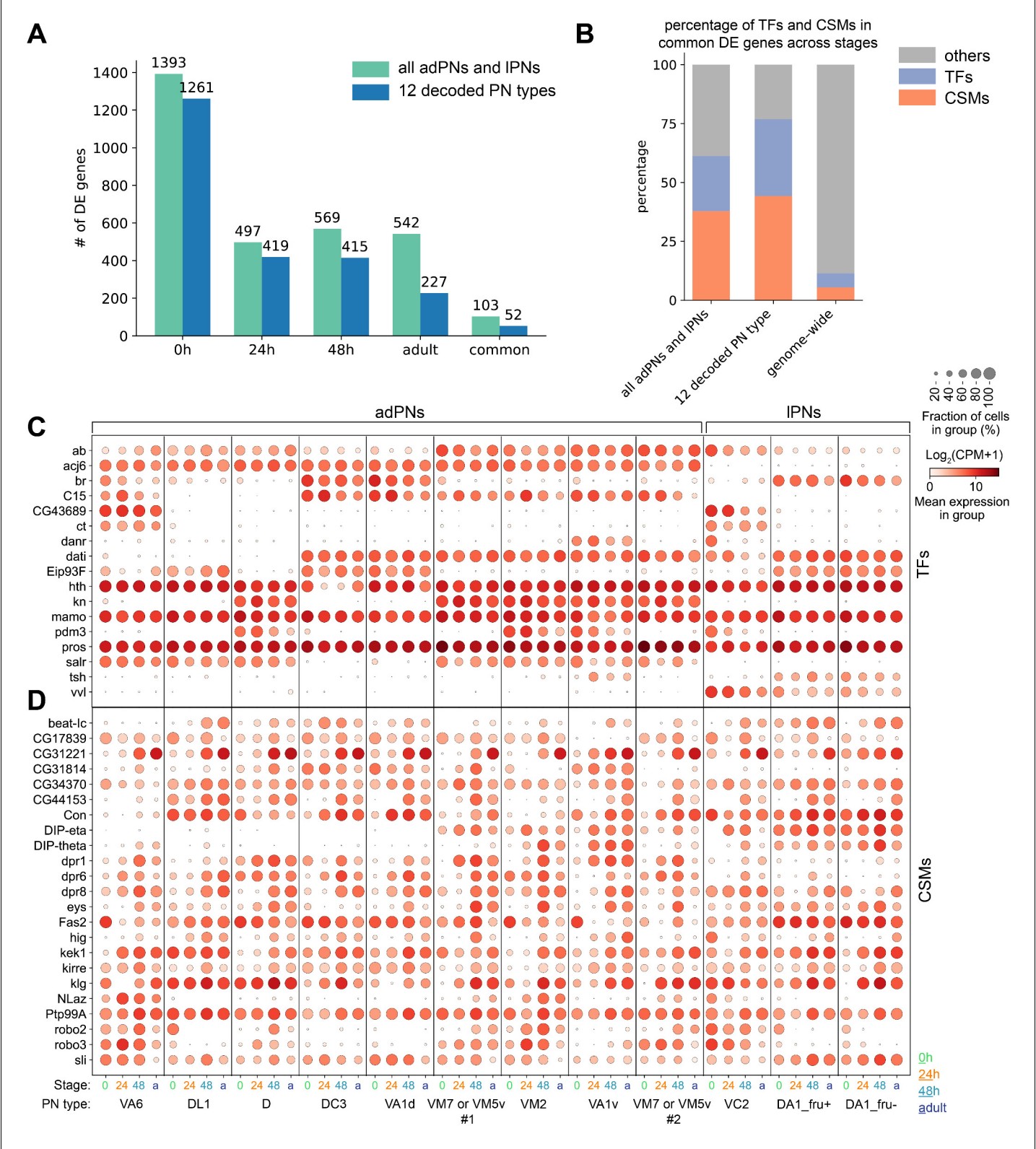

**Figure 9.** Differentially expressed genes between different projection neuron (PN) types. (**A**) Number of differentially expressed (DE) genes identified at each developmental stage among all excitatory PN clusters or among the 12 PN types that are matched in all four stages. A total of 103 and 52 genes are differentially expressed in all four stages among all excitatory PN types or among the 12 PN types, respectively. (**B**) Percentage of transcription factors (TFs) or cell-surface molecules (CSMs) from the list of genes that are differentially expressed among PNs in all four stages

*Figure 9 continued on next page*

*Figure 9 continued*

compared to the genome-wide percentage. (**C, D**) Dot plot of the 17 TFs (**C**) and 23 CSMs (**D**) that are differentially expressed in all four stages among the 12 PN types.

The online version of this article includes the following figure supplement(s) for figure 9:

**Figure supplement 1.** Dot plot of 114 TFs that are differentially expressed in any of the four stages among the 12 projection neuron (PN) type matched across all stages.

**Figure supplement 2.** Dot plot of 228 CSMs that are differentially expressed in any of the four stages among the 12 projection neuron (PN) type matched across all stages.

## Discussion

### Deciphering single-cell transcriptomes for connectivity-defined neuronal types

Traditionally, neurons are classified based on their morphology, physiology, connectivity, and signature molecular markers. More recently, scRNA-seq has allowed classification of cell types based entirely on their transcriptomes. Many studies have illustrated that cell-type classification based on the single-cell transcriptomes largely agrees with classifications by some of the more traditional criteria (*Zeng and Sanes, 2017*).

For *Drosophila* olfactory PNs, the most prominent type-specific feature is their pre- and post-synaptic connections, which determines their olfactory response profiles and the higher order neurons they relay olfactory information to. Thus, different PN types are largely defined by the differences in their connectivity. We have previously observed that the transcriptomic identity of PNs corresponds well with their types during development, and for three identified PN types, transcriptomic differences peak during the circuit assembly stage (*Li et al., 2017*). Here, we generalized these findings across many more PN types by showing that transcriptomic differences are the highest around 24 hr APF, a stage when PNs are making wiring decisions and preparing cues for subsequent ORN–PN matching (*Figure 4*), and by demonstrating that clustering of PNs according to their types from all stages are best done using differentially expressed genes at 24 hr APF (*Figure 5*). Additionally, our data indicate that at certain stages, differences among those type-specific genes can be masked by other genes belonging to pathways of a more dominating biological process (such as neurite pruning at 0 hr APF for PNs). As a consequence, it may be challenging to identify genes carrying type-specific information at certain timepoints even when sophisticated algorithms are applied, which can lead to underestimation of cell type diversity. Our observation of peaked transcriptome diversity in developing PNs has also been observed in the *Drosophila* optic lobe recently (*Özel et al., 2021*). Thus, in order to accurately classify single-cell transcriptomes, especially for connectivity-defined neuronal types such as fly olfactory PNs, it may be a general strategy to first obtain their single-cell transcriptomes during circuit assembly and then use this information to supervise cell-type classification in other developmental stages, including adults.

### Tracing the same cell type in different states

Both cell types and their biological states can split single-cell transcriptomes into distinct clusters (*Zeng and Sanes, 2017*; *Cembrowski and Menon, 2018*; *Tasic, 2018*). We observed that the same PN types of different developmental stages—reflecting different states—indeed exhibit very distinct transcriptomic profiles (*Figures 5* and *6*). To identify transcriptomic clusters corresponding to the same PN types across multiple timepoints, we developed and applied two complementary methods—one manual based on the marker gene expression, and one automatic based on the similarity between transcriptomic clusters. By applying both methods, we can confidently track the transcriptomes of the same cell type throughout development and study the unique molecular features of each stage. We note that two other methods for tracing transcriptomes of the same neuronal types across development—batch-correction to cluster same cell types across different stages, and training an artificial neural network to classify cell type—have been applied successfully in recent single-cell transcriptome studies of cells in the developing *Drosophila* optic lobe (*Kurmangaliyev et al., 2020*; *Özel et al., 2021*).

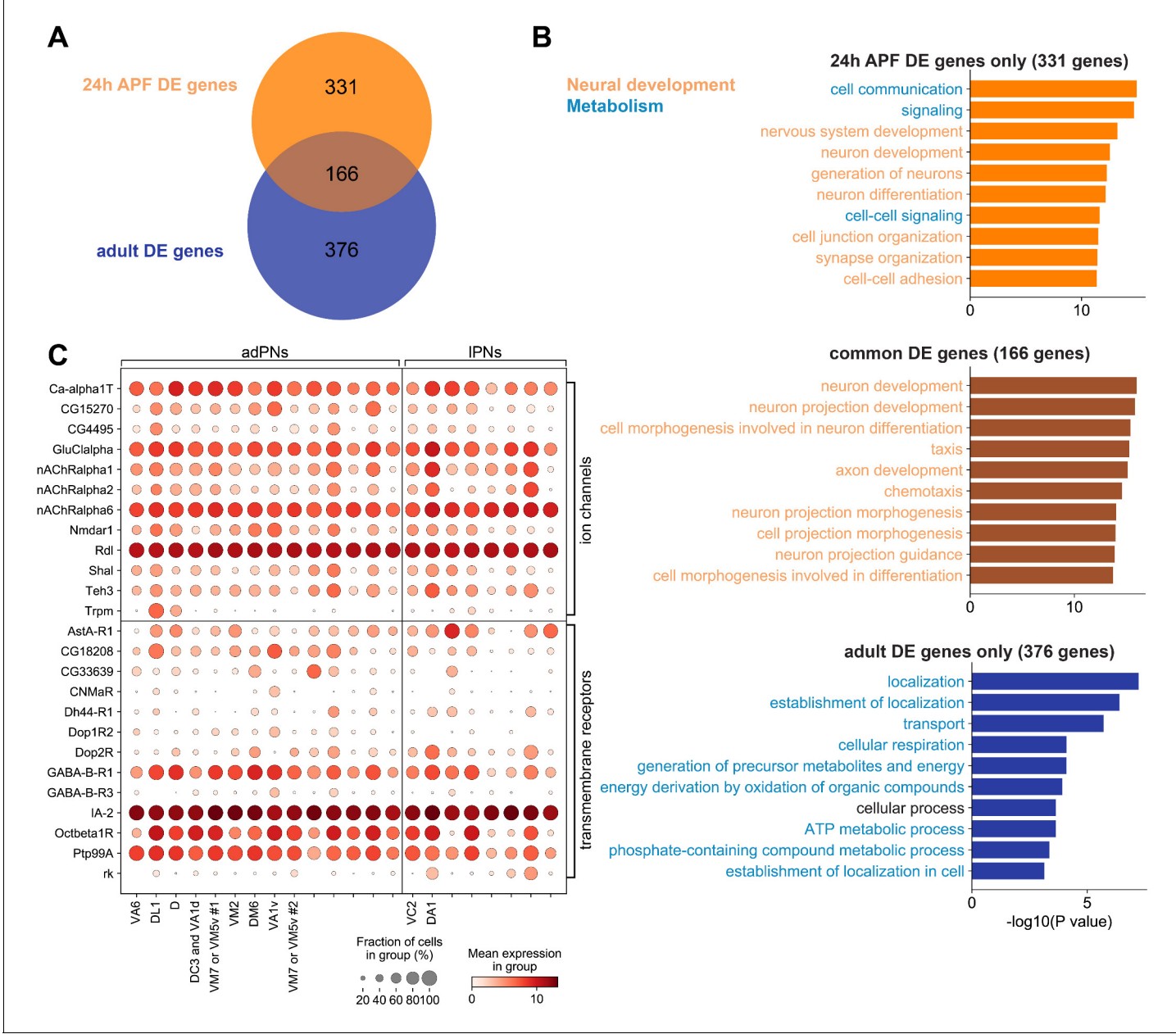

**Figure 10.** Differentially expressed genes among different projection neuron (PN) types in the adult stage. (A) Venn diagram of differentially DE genes at 24 hr APF (497 genes) and in adults (542 genes). (B) Top 10 biological process terms of DE genes found in 24 hr APF PNs only (top), in both 24 hr APF and adults PNs (middle), and in adult PNs only (bottom). GO terms associated with neural development are colored in orange, and GO terms associated with metabolism are colored in blue. (C) Dot plot of adult DE genes that belong to the 'ion channels' (top) or 'transmembrane receptors' (bottom) gene group from FlyBase. PN types are separated by lineage and decoded PN types are labeled and ordered according to their birth order within each lineage.

Together, those methods can be applied to other single-cell studies where diverse cell types and multiple states are involved. Those methods can be especially useful for tissues with high cellular diversity but lack unique markers for each cell type.

## Using single-cell RNAseq data to identify new candidate molecules for future studies

In this study, we have obtained high-quality single-cell transcriptomes of most excitatory PNs from early pupal stage to adulthood (*Figure 1*). We have used combinations of markers and drivers to

decode the transcriptomic identity of 21 transcriptomic clusters at 24 hr APF (*Figure 2*), and matched clusters representing the same PN type across four developmental stages (*Figure 7*).

Using this rich and well-annotated dataset, researchers can now explore different aspects of PN development and function to identify candidate molecules for future studies. For example, one can search for novel molecules involved in neurite pruning among the differentially expressed genes between the embryonically-born and larval-born PNs at 0 hr APF (*Figure 4B–D*). Developmentally enriched genes and genes that are differentially expressed among different PN types, on the other hand, can be good candidates for studies on neural development and wiring specificity (*Figure 3* and *Figure 9*). Differentially expressed neuronal signaling genes in adult PNs can be used to explore differences in physiological properties and information processing (*Figure 10*). In addition, driver lines for specific types of PNs can be made using genes that show consistent expression pattern across different stages (*Figure 7—figure supplement 2*) to label and genetically manipulate specific PN types. Together with several recent in depth scRNAseq studies of cells in the visual and olfactory system across multiple stages (*Jain et al., 2020*; *Kurmangaliyev et al., 2020*; *McLaughlin et al., 2021*; *Özel et al., 2021*), these studies have established foundations of gene expression for *Drosophila* olfactory and visual systems and should catalyze new biological discoveries.

## Materials and methods

**Key resources table**

| Reagent type (species) or resource | Designation | Source or reference | Identifiers | Additional information |
|---|---|---|---|---|
| Genetic reagent (*D. melanogaster*) | GH146-GAL4 | *Stocker et al., 1997* | RRID:BDSC_30026 | |
| Genetic reagent (*D. melanogaster*) | VT033006-GAL4 | *Tirian and Dickson, 2017* | RRID:BDSC_73333 | |
| Genetic reagent (*D. melanogaster*) | Mz19-GAL4 | *Jefferis et al., 2004* | RRID:BDSC_41573 | |
| Genetic reagent (*D. melanogaster*) | knot-GAL4 | *Lee et al., 2018* | RRID:BDSC_67516 | |
| Genetic reagent (*D. melanogaster*) | split#28 GAL4 | Yoshi Aso (unpublished) | N/A | SS01265 |
| Genetic reagent (*D. melanogaster*) | split#7 GAL4 | Yoshi Aso (unpublished) | N/A | SS01867 |
| Genetic reagent (*D. melanogaster*) | split#15 GAL4 | Yoshi Aso (unpublished) | N/A | SS01165 |
| Genetic reagent (*D. melanogaster*) | GH146-Flp | *Hong et al., 2009* | N/A | |
| Genetic reagent (*D. melanogaster*) | UAS-FRT-STOP-FRT-mCD8GFP | *Hong et al., 2009* | RRID:BDSC_30125 | |
| Genetic reagent (*D. melanogaster*) | zfh2-GAL4 | *Lee et al., 2018* | RRID:BDSC_86479 | |
| Genetic reagent (*D. melanogaster*) | Act-FRT-STOP-FRT-GAL4 | *Pignoni and Zipursky, 1997* | N/A | |
| Genetic reagent (*D. melanogaster*) | UAS-Flp | *Duffy et al., 1998* | N/A | |
| Genetic reagent (*D. melanogaster*) | C15-p65$^{AD}$ | *Xie et al., 2019* | N/A | |
| Genetic reagent (*D. melanogaster*) | C15-GAL4$^{DBD}$ | This study | N/A | |
| Genetic reagent (*D. melanogaster*) | danr-P65$^{AD}$ | This study | N/A | |
| Genetic reagent (*D. melanogaster*) | VT033006-GAL4$^{DBD}$ | Yoshi Aso (unpublished) | N/A | |
| Genetic reagent (*D. melanogaster*) | DIP-zeta-GAL4 | *Cosmanescu et al., 2018* | RRID:BDSC_90317 | |
| Genetic reagent (*D. melanogaster*) | DIP-eta-GAL4 | *Cosmanescu et al., 2018* | RRID:BDSC_90318 | |
| Genetic reagent (*D. melanogaster*) | AstA-GAL4 | *Deng et al., 2019* | RRID:BDSC_84593 | |
| Genetic reagent (*D. melanogaster*) | DIP-beta-GAL4 | *Carrillo et al., 2015* | RRID:BDSC_90316 | |
| Genetic reagent (*D. melanogaster*) | kn-GAL4$^{DBD}$ | This study | N/A | |

*Continued on next page*

*Continued*

| Reagent type (species) or resource | Designation | Source or reference | Identifiers | Additional information |
|---|---|---|---|---|
| Genetic reagent (*D. melanogaster*) | *elav-GAL4$^{DBD}$* | *Luan et al., 2006* | N/A | |
| Antibody | Rat monoclonal anti-Ncad | Developmental Studies Hybridoma Bank | RRID:AB_528121 | (1:40 in 5% normal goat serum) |
| Antibody | Chicken polyclonal anti-GFP | Aves Labs | RRID:AB_10000240 | (1:1000 in 5% normal goat serum) |
| Software, algorithm | ZEN | Carl Zeiss | RRID:SCR_013672 | |
| Software, algorithm | ImageJ | National Institutes of Health | RRID:SCR_003070 | |
| Software, algorithm | Illustrator | Adobe | RRID:SCR_010279 | |
| Software, algorithm | STAR 2.5.4 | *Dobin et al., 2013* | RRID:SCR_015899 | https://github.com/alexdobin/STAR |
| Software, algorithm | HTseq 0.11.2 | *Anders et al., 2015* | RRID:SCR_005514 | https://github.com/htseq/htseq |
| Software, algorithm | Scanpy | *Wolf et al., 2018* | RRID:SCR_018139 | https://scanpy.readthedocs.io/en/stable/ |
| Software, algorithm | Iterative Clustering for Identifying Markers (ICIM) | *Li et al., 2017* | N/A | https://github.com/felixhorns/FlyPN |
| Recombinant DNA reagent | *pT-GEM(0)* (plasmid) | *Diao et al., 2015* | RRID:Addgene_62891 | |
| Recombinant DNA reagent | *pBS-KS-attB2-SA(2)-T2A-Gal4DBD-Hsp70* (plasmid) | *Diao et al., 2015* | RRID:Addgene_62904 | |
| Recombinant DNA reagent | *pU6-BbsI-chiRNA* (plasmid) | *Gratz et al., 2013* | RRID:Addgene_45946 | |

## *Drosophila* stocks and genotypes

Flies are maintained on standard cornmeal medium at 25°C with 12 hr light–dark cycle. The following lines were used in this study: *GH146-GAL4* (*Stocker et al., 1997*), *VT033006-GAL4* (*Tirian and Dickson, 2017*), *Mz19-GAL4* (*Jefferis et al., 2004*), *knot-GAL4* (*Lee et al., 2018*), *GH146-Flp, UAS-FRT-STOP-FRT-mCD8-GFP* (*Potter et al., 2010*), *zfh2-GAL4* (*Lee et al., 2018*), *Act-FRT-STOP-FRT-GAL4* (*Pignoni and Zipursky, 1997*), *UAS-Flp* (*Duffy et al., 1998*), *C15-p65$^{AD}$* (*Xie et al., 2019*), *DIP-beta-GAL4, DIP-eta-GAL4, DIP-zeta-GAL4* (*Carrillo et al., 2015*; *Cosmanescu et al., 2018*), *AstA-GAL4* (*Deng et al., 2019*), and *elav-GAL4$^{DBD}$* (*Luan et al., 2006*). *VT033006-GAL4$^{DBD}$*, split-GAL4 line #7 (SS01867), #15 (SS01165), and #28 (SS01265) are unpublished reagents generously provided by Yoshi Aso (Janelia Research Campus).

*danr-p65$^{AD}$* was generated using CRISPR mediated knock-in. Approximately 2000 bp of genomic sequence flanking the targeted insertion site was amplified by Q5 hot-start high-fidelity DNA polymerase (New England Biolabs) and inserted into *pCR-Blunt-TOPO* vectors (Thermo Fisher). Using this vector, we generated homology-directed repair (HDR) vector *TOPO-danr-T2A-p65AD-P3-RFP* by inserting *T2A-p65(AD)::Zip+* and *3XP3-RFP-SV40* (cloned from *pT-GEM(0)* Addgene #62891) 45 bp downstream of the start codon of *danr*. CRISPR guide RNA (gRNA) targeting a sequence inside *danr* (AACATCCGGATGAGCACGCG) were designed by the flyCRISPR Target Finder tool and cloned into a *pU6-BbsI-chiRNA* vector (Addgene #45946). The HDR and gRNA vectors were co-injected into *nos-Cas9* (gift from Dr. Ben White) embryos. RFP+ progenies were selected and individually balanced.

*kn-GAL4$^{DBD}$* was generated by co-injecting *pBS-KS-attB2-SA(2)-T2A-GAL4DBD-Hsp70* (Addgene #62904) and ΦC31 into the embryos of *MI15480* (BL61064). All *yellow$^-$* progenies were individually balanced.

C15-GAL4$^{DBD}$ was generated using methods similar to *danr-p65$^{AD}$*. But because C15 have been shown to be involved in PN dendrite targeting (*Li et al., 2017*), instead of inserting driver elements into the coding region, the stop codon of *C15* was replaced by *T2A-GAL4(DBD)::Zip+* to prevent disruption of the gene.

## Immunofluorescence

Fly brains were dissected and immunostained according to previously described methods (*Wu and Luo, 2006*). Primary antibodies used in this study included rat anti-Ncad (N-Ex #8; 1:40; Developmental Studies Hybridoma Bank), chicken anti-GFP (1:1000; Aves Labs). Secondary antibodies conjugated to Alexa Fluor 488/647 (Jackson ImmunoResearch) were used at 1:250. Five percent normal goat serum in phosphate buffered saline was used for blocking and diluting antibodies. Confocal images were collected with a Zeiss LSM 780 and processed with ImageJ.

## Single-cell RNA sequencing procedure

Single-cell RNA sequencing was performed following previously described protocol (*Li et al., 2017*). Briefly, *Drosophila* brains with mCD8-GFP labeled cells using specific GAL4 drivers were dissected at appropriate timepoints (0–6 hr APF, 24–30 hr APF, 48–54 hr APF, and 1–5 day adults). Optic lobes were removed from brain during dissection for all timepoints except for 0–6 hr APF. Single-cell suspension were prepared and GFP positive cells were sorted using Fluorescence Activated Cell Sorting (FACS) into individual wells of 384-well plates containing lysis buffer using SH800 (Sony Biotechnology). Full-length poly(A)-tailed RNA was reverse-transcribed and amplified by PCR following the SMART-seq2 protocol (*Picelli et al., 2014*). cDNA was digested using lambda exonuclease (New England Biolabs) and then amplified for 25 cycles. Sequencing libraries were prepared from amplified cDNA, pooled, and quantified using BioAnalyser (Agilent). Sequencing was performed using the Novaseq 6000 Sequencing system (Illumina) with 100 paired-end reads and 2 × 8 bp index reads.

## Quantification and statistical analysis

Unless otherwise specified, all data analysis was performed in Python using Scanpy (*Wolf et al., 2018*), Numpy, Scipy, Pandas, scikit-learn, and custom single-cell RNA-seq modules (*Li et al., 2017*; *Brbić et al., 2020*). Gene Ontology analysis were performed using Flymine (*Lyne et al., 2007*). Sequencing reads and preprocessed sequence data are available in the NCBI Gene Expression Omnibus (GSE161228). Custom analysis code is available at https://github.com/Qijing-Xie/FlyPN_development.

## Sequence alignment and preprocessing

Reads were aligned to the *Drosophila melanogaster* genome (r6.10) using STAR (2.5.4) (*Dobin et al., 2013*). Gene counts were produced using HTseq (0.11.2) with default settings except ''-m intersection-strict' (*Anders et al., 2015*). We removed low-quality cells having fewer than 100,000 uniquely mapped reads. To normalize for differences in sequencing depth across individual cells, we rescaled gene counts to counts per million reads (CPM). All analyses were performed after converting gene counts to logarithmic space via the transformation $Log_2(CPM+1)$. We further filter out non-neuronal cells by selecting cells with high expression of canonical neuronal genes ( *elav*, *brp*, *Syt1*, *nSyb*, *CadN*, and *mCD8-GFP*). We retained cells expressing at least 8 $Log_2(CPM+1)$ for least 2/6 markers.

## Dimensionality reduction and clustering

To select variable genes for dimensionality reduction, we used previously described methods to search for either overdispersed genes (*Satija et al., 2015*) or ICIM genes (*Li et al., 2017*). We then further reduced dimensionality using tSNE to project the reduced gene expression matrix into a two-dimensional space (*van der Maaten and Hinton, 2008*). We observed that most of our recently sequenced cells using NovaSeq exhibited some small batch effect with PNs sequenced using Next-Seq [PNs from *Li et al., 2017*]. To overcome this batch effect (in *Figure 2*, and *Figure 7—figure supplement 2A,C*), we performed principal component analysis (PCA) on the ICIM matrix, applied Harmony to correct for batch effect on the principal components (PCs) (*Korsunsky et al., 2019*), and used tSNE to further project the Harmony-corrected PCs into a two-dimensional space.

To cluster PNs in an unbiased manner, we applied the hierarchical density-based clustering algorithm, HDBSCAN, on the tSNE projection (*McInnes et al., 2017*). Parameters min_cluster_size and min_samples were adjusted to separate clusters representing different types of PNs. In addition, we also clustered cells using an independent, community-detection method called Leiden on the neighborhood graph computed based on the ICIM gene matrix (*Blondel et al., 2008*; *Levine et al., 2015*; *McInnes et al., 2018*). Both methods appeared to agree with each other for all datasets we examined (examples in *Figure 5—figure supplement 1*), and we assigned PN types in *Figure 2* based on HDBSCAN clustering.

## Global level dynamic gene identification

To identify dynamically expressed genes on the global level (*Figure 3*), we first identified the top 150 most differentially expressed genes (Mann-Whitney U test) between every two stages and combined them to obtain a set of 474 dynamic genes. We calculated the median expression of each gene at each timepoint and normalized these median expression values by dividing them by the maximum value across time points. We then performed dimensionality reduction on the expression profiles of the genes using tSNE, and identified clusters using HDBSCAN on the projected coordinates. This resulted in identification of eight sets of genes with distinct dynamic profiles, of which two sets are upregulated (*Figure 3E*), four sets are down regulated (*Figure 3D*), and two sets without obvious trend from 0 hr APF to adult cells (data not shown).

## Transcriptomic similarity calculation

To analyze the transcriptome differences of PNs in different stages (*Figure 4E,F*), we first isolated lPNs and adPNs to analyze cells from each lineage separately. Cell-level analysis was performed by calculating for each cell mean inverse Euclidean distance in the two-dimensional UMAP space from all other cells within each stage using the 1215 genes identified by ICIM from most PNs of all stages (*Figure 3A*). Box plots show the distance distribution at each stage (*Figure 4E and F*, left). Cluster-level analysis was performed on the MARS clusters. We identified a set of differentially expressed genes for each cluster and calculated Pearson correlation on differentially expressed genes between all pairs of clusters. Bar plots represent mean values across all pairs and errors are 95% confidence intervals determined by bootstrapping with n = 1000 iterations (*Figure 4E and F*, right).

## PN type identification for most PNs

We observed that the transcriptomes of different PN types are the most distinct at 24 hr APF and variable genes identified at this stage carry type-specific information (*Figure 5*). Therefore, we calculated the differentially expressed genes among 24 hr APF clusters and applied MARS to identify clusters in the space of those genes. MARS is able to reuse annotated single-cell datasets to learn shared low-dimensional space of both annotated and unannotated datasets in which cells are grouped according to their cell types. However, initially we did not have any annotated experiments, so we first applied MARS to annotate 24 hr APF clusters. We then used 24 hr APF clusters as annotated dataset and moved to annotate PNs at 48 hr APF. We then repeated the same procedure by gradually increasing our set of annotated datasets. In particular, we used 24 hr and 48 hr APF data to help in annotating 0 hr APF, and finally all three datasets (0 hr, 24 hr, 48 hr) for the adult PNs. We proceed in this order according to the expected difficulty to identify PN types at a particular stage (*Figure 5*). At each stage, we ran MARS multiple times with different random initializations and architecture parameters to increase our confidence in the discovered clusters, and combined annotations from these different runs. For each cluster, we additionally manually checked the expressions of known PN markers to confirm the annotations.

## Matching clusters representing the same PN type across development using marker expression

For each cluster, we used Mann-Whitney U test to find genes that are highly expressed in that cluster compared to the rest. Then, among those genes, we searched for genes or two-gene combinations which are uniquely expressed in one cluster. We check each gene or combination of genes at the other stages, and if they are also only expressed in one cluster and they are of the same lineage,

we consider them to be the same types of PNs. Genes used to match clusters representing the same PN types at different timepoints are summarized in a dot-plot in *Figure 7—figure supplement 2*.

In addition, we used previously sequenced subset of PNs using *Mz19-GAL4* and *kn-GAL4* to overlay with most PNs in combinations of those markers to confirm our matching.

## Matching clusters representing the same PN type across development using similarity calculation

For each cluster, we found the set of differentially expressed genes in that cluster compared to all other clusters at the same stage. Next, we computed the similarity of the sets of identified differentially expressed genes between all pairs of clusters across subsequent stages. Specifically, we computed similarity scores between all pairs of clusters from (i) 0 hr and 24 hr APF, (ii) 24 hr and 48 hr APF, and (iii) 48 hr and adult APF. The similarity of the sets of differentially expressed genes was computed as the Jaccard similarity index defined as the ratio of the cardinality of the intersection of two sets and the cardinality of the union of the sets. We excluded clusters representing vPNs and APLs for matching most PNs across four stages (*Figure 7*). For each cluster, we then identified its most similar cluster at the adjacent stage according to the Jaccard index. If the clusters between two stages coincide—meaning that two clusters from two stages have the highest similarity to each other, we consider the clusters to be matched. Empirically, we found this matching procedure to be stringent, resulting in high confidence matching pairs.

## Correlation between different PN types

MARS clusters of excitatory PNs were used for analysis in *Figure 8*. We performed PCA on the entire matrix and calculated their correlation based on the PCs. Dendrograms shown in *Figure 8—figure supplement 1* are generated using distance calculated using Farthest Point Algorithm and organized so the distance between successive leaves is minimal.

To observe the relationship between birth timing and their transcriptomic similarity, for each stage, we selected adPN clusters, performed PCA among all genes detected, calculated their correlation, and plotted the correlation matrices according to their birth order (*Yu et al., 2010*; *Figure 8B*). For the two clusters representing either VM7 or VM5v PNs, we ordered them based on their correlation with decoded PN types whose birth order are adjacent to either of these two PN types. We are showing adPNs in the figure because we decoded much fewer transcriptomic clusters belonging to the lPN lineage, which is too few to carry out analysis shown in *Figure 8C–D* with robust statistical backing. Nevertheless, we still observed higher correlation between lPN types with adjacent birth-order in 0 hr and 24 hr APF (data not shown).

## Spearman's rank correlation calculation and permutation test

For consistency, eight adPN types that were decoded across four stages were selected for this analysis (*Figure 8C*). For each PN type X, the group of PNs that are born either earlier or later than X was selected depending on which direction contains more PN types (each group contains at least five types of PNs). Then, we ranked the PN types according to their correlation with X and calculated the Spearman's rank correlation of this ranking with the ranking based on their birth order. For each stage, we obtained the average correlation coefficients and plotted the result as a red dot on the x-axis for each timepoint. Higher value indicates higher correlation between birth order and order calculated based on their transcriptomic similarity.

To determine if we can reject the null hypothesis that the adPN transcriptomic similarity do not covary with the ranks of the birth order, we performed permutation test. We randomly shuffled the birth order and performed the aforementioned correlation calculation for 5000 iterations. The distribution of the simulated average correlations is shown in the histogram of *Figure 8C*. We obtained the p-value by dividing the number of times of the simulated correlation is greater than the observed correlation by the total number of iterations.

## Developmental trajectory analysis

Pseudo-time analysis of 0 hr APF adPNs was performed using the monocle package in R (*Trapnell et al., 2014*; *Qiu et al., 2017*; *Cao et al., 2019*). We selected only adPNs born at larval stage because the embryonically born adPNs have a very distinct transcriptomes which skew

clustering. We applied the dimensionality reduction method UMAP (*Becht et al., 2019*) on 561 24 hr ICIM genes to resolve distinct PN types. This dimensionally reduced dataset was then used as the basis for a developmental trajectory graph created by Monocle 3. We then selected the cluster representing DL1 PNs to be the root node of the trajectory and computed the pseudo-times based on distance from the root in accordance to the trajectory.

## Differential gene expression analysis

We used adPN and lPN clusters to identify differentially expressed genes at each stage (*Figure 9*). We performed Mann-Whitney U test on each cluster compared to the rest of the clusters at each developmental stage and applied Benjamini-Hochberg Procedure to adjust p-value. Genes with an adjusted p-value of less than 0.01 were kept for our analysis.

To identify genes that are transcription factors (TFs), cell surface molecules (CSM), ion channels, and transmembrane receptors, we used curated lists. The TF list was from the FlyTF database (*Pfreundt et al., 2010*) and the CSM list was from *Kurusu et al., 2008*. These lists were manually curated to remove spurious annotations and redundancies according to Flybase annotation. Lists of ion channels and transmembrane receptors were based on gene groups obtained from FlyBase. To avoid redundancy, ion channels that also belong to the transmembrane receptor gene group are not plotted as transmembrane receptors (*Figure 9C*, bottom).

## Acknowledgements

We thank Yoshi Aso, Gerald Rubin, Hugo Bellen, Kai Zinn, and Larry Zipursky for the kind gifts of reagents. We thank the Bloomington *Drosophila* Stock Center and the Vienna *Drosophila* Resource Center for fly lines, and Addgene for plasmids. We thank Tom Clandinin, Yanyang Ge, Julia Kaltschmidt, Justus Kebschull, Kang Shen, Andrew Shuster, and all Luo lab members for technical support and insightful advice on this study. We thank Mary Molacavage for administrative assistance.

## Additional information

### Funding

| Funder | Grant reference number | Author |
| --- | --- | --- |
| National Institutes of Health | R01 DC005982 | Liqun Luo |
| National Institutes of Health | 1K99AG062746 | Hongjie Li |
| Howard Hughes Medical Institute | Investigator | Liqun Luo |
| Bertarelli Foundation | Bertarelli Fellow | Qijing Xie |
| Wu Tsai Neurosciences Institute | Interdisciplinary Postdoctoral Scholar | Hongjie Li |
| Wu Tsai Neurosciences Institute | Neuro-omics program | Liqun Luo Stephen R Quake Jure Leskovec |
| Chan Zuckerberg Biohub | Investigator | Jure Leskovec Stephen R Quake |

The funders had no role in study design, data collection and interpretation, or the decision to submit the work for publication.

### Author contributions

Qijing Xie, Conceptualization, Resources, Data curation, Software, Formal analysis, Validation, Investigation, Visualization, Methodology, Writing - original draft, Writing - review and editing; Maria Brbic, Resources, Data curation, Software, Formal analysis, Visualization, Methodology, Writing - review and editing; Felix Horns, Sai Saroja Kolluru, Robert C Jones, Jiefu Li, Anay R Reddy, Zhuoran Li, Colleen N McLaughlin, Tongchao Li, Chuanyun Xu, David Vacek, David J Luginbuhl, Jure Leskovec, Resources; Anthony Xie, Sayeh Kohani, Formal analysis; Stephen R Quake, Resources, Funding

acquisition; Liqun Luo, Conceptualization, Resources, Supervision, Funding acquisition, Writing - original draft, Writing - review and editing; Hongjie Li, Conceptualization, Resources, Data curation, Formal analysis, Supervision, Investigation, Methodology, Writing - review and editing

**Author ORCIDs**
Qijing Xie (iD) https://orcid.org/0000-0002-0997-8326
Felix Horns (iD) http://orcid.org/0000-0001-5872-5061
Robert C Jones (iD) http://orcid.org/0000-0001-7235-9854
Jiefu Li (iD) http://orcid.org/0000-0002-0062-4652
Stephen R Quake (iD) http://orcid.org/0000-0002-1613-0809
Liqun Luo (iD) https://orcid.org/0000-0001-5467-9264

**Decision letter and Author response**
Decision letter https://doi.org/10.7554/eLife.63450.sa1
Author response https://doi.org/10.7554/eLife.63450.sa2

## Additional files

### Supplementary files

• Transparent reporting form

### Data availability

Raw sequencing reads and preprocessed sequence data have been deposited in GEO under accession code GSE161228.

The following dataset was generated:

| Author(s) | Year | Dataset title | Dataset URL | Database and Identifier |
|---|---|---|---|---|
| Xie Q | 2020 | Temporal evolution of single-cell transcriptomes of Drosophila olfactory projection neurons | https://www.ncbi.nlm.nih.gov/geo/query/acc.cgi?acc=GSE161228 | NCBI Gene Expression Omnibus, GSE161228 |

The following previously published dataset was used:

| Author(s) | Year | Dataset title | Dataset URL | Database and Identifier |
|---|---|---|---|---|
| Horns F | 2017 | Classifying Drosophila Olfactory Projection Neuron Subtypes by Single-cell RNA Sequencing | https://www.ncbi.nlm.nih.gov/geo/query/acc.cgi?acc=GSE100058 | NCBI Gene Expression Omnibus, GSE100058 |

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
