## [Decision Letter]

**Acceptance summary:**

Your study sets a fine example of high-resolution decoding of transcriptional identity using a combination of single-cell transcriptomics, genetic driver lines, and imaging. It will be a useful resources for the community!

**Decision letter after peer review:**

Thank you for submitting your article "Temporal evolution of single-cell transcriptomes of *Drosophila* olfactory projection neurons" for consideration by *eLife*. Your article has been reviewed by three peer reviewers, and the evaluation has been overseen by a Reviewing Editor and K VijayRaghavan as the Senior Editor. The following individual involved in review of your submission has agreed to reveal their identity: Scott Barish (Reviewer #2).

The reviewers have discussed the reviews with one another and the Reviewing Editor has drafted this decision to help you prepare a revised submission.

This paper from the Luo lab builds upon their previous paper on single cell RNA profiling from developing projection neuron subpopulations in the *Drosophila* olfactory system. In this paper, the authors expanded their sequence analysis to new populations subsets for projection neuron (PN) classes using new combinations of split GAL4 lines to label new projection neuron subpopulations for cell sorting and sequencing. Each PN class generally innervates a single antennal lobe glomerulus to make connections with a single class of olfactory receptor neurons (ORNs). Different PN classes are born from 3 types of neuroblast lineages in a specific temporal order that form anterodorsal, lateral PN clusters that represent excitatory cholinergic and ventral inhibitory GABAergic PN clusters. Incorporating this new dataset with their previous scRNAseq data, they were able to increase the resolution of the sequence clusters representing different PNs during development, identifying 20 PN classes. In addition, comparing sequence data for each developmental time point, the authors were able to provide developmental trajectories for PN transcriptional profiles and how they change over time. The sequence analysis shows that the transcriptional profiles across PNs are show the highest observable diversity in early to mid-pupal stages when the PNs are differentiating and connecting with appropriate ORN classes within glomeruli. However, this transcriptional diversity across PN populations is dissolved by adult stages in mature PN populations. The authors were also able to show that PNs that are in the same lineage with adjacent birth order were transcriptionally more similar. Even though this paper provides an incremental progress from the previous study from the Luo lab on PN transcriptional profiles during development, it does provide some additional information on how transcriptional diversity and developmental trajectories are influenced by lineage, birth order and birth timing of diverse PN populations during development. The study sets a fine example of high-resolution decoding of transcriptional identity using a combination of single-cell transcriptomics, genetic driver lines, and imaging. A few points to address or add to the manuscript are listed below before the paper can be published in e*Life*.

1) Some of the wording on the figures are extremely small. Please make them.

2) From Materials and methods, it seems the sequence data and the codes have not been submitted to the public databases yet. Please make sure these are uploaded.

3) The clusters in Figure 2C, F, and J where the new sequence data is overlayed onto the *GH146+* PN data at 24 hours: I am not sure why the gray *GH146+* cluster patterns appear different in each tSNE graph. Aren't they supposed to be the same? They appear very similar in Figure 2—figure supplement 1.

4) Given that this is a follow up study, It would be good to see as much data as possible that can provide new knowledge about the transcriptional profiles. Could the authors provide a list and a heatmap matrix for PN cluster specific expression of some key gene families like cell surface molecules, transcription factors, neurotransmitter receptors, and ion channels for each developmental time point? Throughout the text there are mentions of these gene groups but it would be good to see it as a figure for each in a supplement.

5) The bioinformatics analyses presented are well explained, and thorough. However, there is an important aspect missing. The 24h cells have the greatest power in clustering, finding all PN subtypes, while the adult is the weakest (transcriptomes converge). Nevertheless, using MARS (alternatives using simple SVM classifiers could have been applied, as benchmarked by Abdelaal et al. 2019, Genome Biology), and followed-up using more in depth analyses, the authors show that the adult cells show similarities to the 24h subtypes. It would be a great added value if the cells could be analysed all together, across time points, using a couple of batch effect removal techniques (Harmony, BBKNN, Scanorama,..). It would follow from the results that the 24h cells would drive the clustering, but that the other time points would co-cluster. This would provide an elegant foundation, finding all subtypes back, with cells from each time point present in each cluster. Next, each subtype can be analysed separately, using trajectory inference, to study the dynamical changes. The current analyses somehow approximate this strategy using an ad hoc combination of methods, which seems reasonable, but would benefit from a comparison with aforementioned batch effect corrections (the batch here would be the time point).

6) A similar study of tracking neuronal subtype development has been carried out for T4/T5 neurons in the optic lobe, as well as other optic lobe subtypes. It would be informative to discuss the current findings in the context of these studies from the Desplan and Zipursky labs.

7) An inference is made to connect developmental trajectories with neuroblast birth order. It seems a missed opportunity to include single-cell transcriptomes of the neuroblasts in this study, for example using scRNA-seq of the larval brain. The authors exploit gene sets from earlier studies – but could the entire data set be used instead? If this is bulk RNA-seq, there are computational techniques to compare them (map them) onto the single-cell data.

8) Ecdysone is mentioned in the manuscript, but there is little investigation into the transcriptome changes that are induced by the ecdysone peak (see also Jain et al., 2020). The dynamic-dynamic and dynamic-stable modules is an intuitive way to identify cell type specific dynamics, but how are these linked to the Ecdyson receptor? Does EcR regulate the same genes in every subtype?

---

## [Author Response]

[…] A few points to address or add to the manuscript are listed below before the paper can be published in eLife.1) Some of the wording on the figures are extremely small. Please make them.

We have adjusted the font size in all figures. For Figure 3B, we are now showing the top 52 deferentially expressed genes instead of top 100 genes to ensure all gene names are big enough to read. The entire list of differentially expressed genes can be found and visualized in the GitHub page.

2) From Materials and methods, it seems the sequence data and the codes have not been submitted to the public databases yet. Please make sure these are uploaded.

We have submitted the raw and processed sequence data to GEO during the review process. All data is available through the Gene Expression Omnibus (GSE161228). Our custom analysis code is available at https://github.com/Qijing-Xie/FlyPN_development. We have added this information in the Materials and methods section.

3) The clusters in Figure 2C, F, and J where the new sequence data is overlayed onto the GH146+ PN data at 24 hours: I am not sure why the gray GH146+ cluster patterns appear different in each tSNE graph. Aren't they supposed to be the same? They appear very similar in Figure 2—figure supplement 1.

In simplified terms, tSNE first put all data randomly in a 2D space and then use the pair-wise distance between all datapoints in a dataset to move similar data closer together. The final position of clusters is dependent on the random initialization as well as every cell in the dataset. Figure 2C and Figure 2—figure supplement 1 are identical (*GH146+* PNs and *Split#28+* PNs).

We realized that this can be confusing for readers not familiar with the details of tSNE. To prevent future confusions, we have changed the wording from “map those cells to *GH146+* PNs” to “plot them with *GH146+* PNs”) and added a sentence explaining why those plots look different in the Figure 2 legend.

4) Given that this is a follow up study, It would be good to see as much data as possible that can provide new knowledge about the transcriptional profiles. Could the authors provide a list and a heatmap matrix for PN cluster specific expression of some key gene families like cell surface molecules, transcription factors, neurotransmitter receptors, and ion channels for each developmental time point? Throughout the text there are mentions of these gene groups but it would be good to see it as a figure for each in a supplement.

We appreciate the reviewers for this suggestion to make our data more accessible. In response to this, we added a new figure (current Figure 9) focused on the differentially expressed genes among PNs at different stages. These analyses revealed that transcription factors (TFs) and cell-surface molecules (CSMs) are highly enriched among differentially expressed genes (Figure 9A, B). We then list TFs and CSMs that are consistently differentially expressed across the four stages in the Figure 9C, D and those that are differentially expressed in any of the four stages in the Figure 9—figure supplements 1 and 2. We added text discussing these findings in a new section entitled “Differentially expressed genes in different PN types.” In addition, we are presenting differentially expressed ion channels, neurotransmitter receptors, GPCRs, and other transmembrane receptors among different adult PN clusters in Figure 10C.

To enhance the flow of the manuscript, we have rearranged the order of the figures by changing the previous Figure 9 (correlation between transcriptomic similarity with birth order) to the current Figure 8, before the description of differentially expressed genes in Figures 9 and 10.

To avoid an excessively long manuscript, we removed the previous Figure 8 and associated text (Gene expression dynamics in a type-specific manner). The data described in the previous Figure 8 is somewhat redundant with the new Figure 9, but the new Figure 9 is more useful and informative to the readers interested in details of differentially expressed genes in different PN types as well as their dynamics.

We thank the reviewers and editors for this suggestion.

5) The bioinformatics analyses presented are well explained, and thorough. However, there is an important aspect missing. The 24h cells have the greatest power in clustering, finding all PN subtypes, while the adult is the weakest (transcriptomes converge). Nevertheless, using MARS (alternatives using simple SVM classifiers could have been applied, as benchmarked by Abdelaal et al. 2019, Genome Biology), and followed-up using more in depth analyses, the authors show that the adult cells show similarities to the 24h subtypes. It would be a great added value if the cells could be analysed all together, across time points, using a couple of batch effect removal techniques (Harmony, BBKNN, Scanorama,..). It would follow from the results that the 24h cells would drive the clustering, but that the other time points would co-cluster. This would provide an elegant foundation, finding all subtypes back, with cells from each time point present in each cluster. Next, each subtype can be analysed separately, using trajectory inference, to study the dynamical changes. The current analyses somehow approximate this strategy using an ad hoc combination of methods, which seems reasonable, but would benefit from a comparison with aforementioned batch effect corrections (the batch here would be the time point).

We thank the reviewers for proposing this elegant approach for analyzing our data. We have indeed tried multiple batch correction methods (including Harmony, BBKNN, and Scanorama) on our data when we first attempted to match same PN type across development. Harmony seems to perform the best on our dataset compared to the other methods we tried. However, we still observed many instances of (1) PNs of the same type at the same stage split into different clusters; (2) PNs of different types merge into the same cluster; (3) no distinguishable cluster formation for many PNs in stages other than 24h APF (Author response image 1)., This has motivated us to take the more laborious approach to develop automatic and manual matching methods described in the manuscript.

Although batch correction methods we tried were unable to fully “correct” changes in the PN transcriptome throughout development, we are aware that these algorithms can be suitable for analyzing other cell types. Therefore, we have added a sentence highlighting this approach along with one other method using neural network classifier in the Discussion section.

**Author response image 1. sa2fig1:** Visualization of most PNs in 4 different developmental stages after batch correction using harmony. Cells are colored based on their developmental stage (left) and the final decoded PN types (right). Circled clusters highlight examples of the three types of issues we described.

6) A similar study of tracking neuronal subtype development has been carried out for T4/T5 neurons in the optic lobe, as well as other optic lobe subtypes. It would be informative to discuss the current findings in the context of these studies from the Desplan and Zipursky labs.

We thank the reviewers for pointing these papers out. Those papers were published after our initial submission therefore we were not able to cite them. In the revised manuscript, we have cited both studies and discussed some of their findings/methods in multiple places of our Results and Discussion sections.

7) An inference is made to connect developmental trajectories with neuroblast birth order. It seems a missed opportunity to include single-cell transcriptomes of the neuroblasts in this study, for example using scRNA-seq of the larval brain. The authors exploit gene sets from earlier studies – but could the entire data set be used instead? If this is bulk RNA-seq, there are computational techniques to compare them (map them) onto the single-cell data.

We thank the reviewers for their enthusiasm for this piece of data. We agreed that the single-cell transcriptomes of PN neuroblast could reveal interesting biological insights. However, it will be a large amount of work as the neuroblast transcriptomes have their own temporal dynamics. We believe it is beyond the scope of this paper, which already contains extensive amount of data.

The reviewers also suggested to compare the bulk RNAseq data of PN neuroblasts with the single-cell RNAseq data of post-mitotic PNs (up to 5 days since their birth). We are unaware of a good analytical method that will allow us to compare different modalities in different cell types of different developmental stage stages. As a matter of fact, Liu et al. found only 63 genes that are differentially expressed in PN neuroblasts of different larval stages from their bulk RNAseq, suggesting that the rest of the transcriptome likely do not carry much birth-order-related information. Therefore, we focused only on these 63 genes for our analysis. As explained in our manuscript, we examined the expression pattern of all 63 genes in PNs (Author response image 1) and found that only 15 of them showed some temporal gradient patterns. We did not include all genes in our manuscript to avoid confusions because many of them are either not very highly expressed in postmitotic PNs or do not have birth-order-related dynamics.

We also edited the paragraph related to this panel to make our conclusions more precise. We hope the reviewers will agree that we accurately described this piece of result and did not make any conclusion that is not supported by the data.

**Author response image 2. sa2fig2:** Expression levels of all 63 PN neuroblasts genes with temporal gradient in 0h APF PNs. Expression levels of 63 genes in adPNs with known identity at 0h APF. These 63 genes are shown to have temporal expression gradient in PN neuroblasts (Liu et al., 2015). The highest expression is normalized as 1 for all genes.

8) Ecdysone is mentioned in the manuscript, but there is little investigation into the transcriptome changes that are induced by the ecdysone peak (see also Jain et al., 2020). The dynamic-dynamic and dynamic-stable modules is an intuitive way to identify cell type specific dynamics, but how are these linked to the Ecdyson receptor? Does EcR regulate the same genes in every subtype?

We thank the reviewers for this observation. The ecdysone pathway plays many important roles in the development of the *Drosophila* brain. While we only discussed its role in the neurite pruning of embryonically born PNs at the beginning of morphogenesis, we also noticed that some genes in this pathway appear in our global dynamic genes. We examine the same TF gene set Jain et al. focused on in lamina neurons and found that some of those genes displayed very similar temporal dynamics in the olfactory PNs (Author response image 3). However, we did not observe any type-specific expression of these genes in PNs besides in the embryonically born PNs at 0h APF. This suggests these TFs in the ecdysone pathway may not regulate the differential gene expression to impact wiring in PNs.

Because the extensive study on ecdysone by Jain et al. hasn’t been published yet, we do not want to discuss the expression dynamics of these genes in PNs, so as not to scoop their novel discoveries.

**Author response image 3. sa2fig3:** Expression of transcription factors of the Ecdysone-induced transcriptional cascade. Heatmap showing the expression of TFs of the Ecdysone-induced transcriptional cascade examined in Jain et al. (note Hr46 is same gene as Hr3).